# LESS IS MORE: RETHINKING FEW-SHOT LEARNING AND RECURRENT NEURAL NETS

## ABSTRACT

The statistical supervised learning framework assumes an input-output set with a joint probability distribution that is reliably represented by the training dataset. The learner is then required to output a prediction rule learned from the training dataset's input-output pairs. In this work, we provide meaningful insights into the asymptotic equipartition property (AEP) (Shannon, 1948) in the context of machine learning, and illuminate some of its potential ramifications for few-shot learning. We provide theoretical guarantees for reliable learning under the information-theoretic AEP, and for the generalization error with respect to the sample size. We then focus on a highly efficient recurrent neural net (RNN) framework and propose a reduced-entropy algorithm for few-shot learning. We also propose a mathematical intuition for the RNN as an approximation of a sparse coding solver. We verify the applicability, robustness, and computational efficiency of the proposed approach with image deblurring and optical coherence tomography (OCT) speckle suppression. Our experimental results demonstrate significant potential for improving learning models' sample efficiency, generalization, and time complexity, that can therefore be leveraged for practical real-time applications.

## 1 INTRODUCTION

In recent years, machine learning (ML) methods have led to many state-of-the-art results, spanning through various fields of knowledge. Nevertheless, a clear theoretical understanding of important aspects of artificial intelligence (AI) is still missing. Furthermore, there are many challenges concerning the deployment and implementation of AI algorithms in practical applications, primarily due to highly extensive computational complexity and insufficient generalization. Concerns have also been raised regarding the effects of energy consumption of training large scale deep learning systems (Strubell et al., 2020). Improving sample efficiency and generalization, and the integration of physical models into ML have been the center of attention and efforts of many in the industrial and academic research community. Over the years significant progress has been made in training large models. Nevertheless, it has not yet been clear what makes a representation good for complex learning systems (Bottou et al., 2007; Vincent et al., 2008; Bengio, 2009; Zhang et al., 2021).

**Main Contributions.** In this work we investigate the theoretical and empirical possibilities of few shot learning and the use of RNNs as a powerful platform given limited ground truth training data. (1) Based on the information-theoretical asymptotic equipartition property (AEP) (Cover & Thomas, 2006), we show that there exists a relatively small set that can empirically represent the input-output data distribution for learning. (2) In light of the theoretical analysis, we promote the use of a compact RNN-based framework, to demonstrate the applicability and efficiency for few-shot learning for natural image deblurring and optical coherence tomography (OCT) speckle suppression. We demonstrate the use of a single image training dataset, that generalizes well, as an analogue to universal source coding with a known dictionary. The method may be applicable to other learning architectures as well as other applications where the signal can be processed locally, such as speech and audio, video, seismic imaging, MRI, ultrasound, natural language processing and more. Training of the proposed framework is extremely time efficient. *Training takes about 1-30 seconds on a GPU workstation and a few minutes on a CPU workstation (2-4 minutes)*, and thus does not require expensive computational resources. (3) We propose an upgraded RNN framework incorporating receptive field normalization (RFN) (see (Pereg et al., 2021), Appendix C)

that decreases the input data distribution entropy, and improves visual quality in noisy environments. (4) We illuminate a possible optimization mechanism behind RNNs. We observe that an RNN can be viewed as a sparse solver starting from an initial condition based on the previous time step. The proposed interpretation can be viewed as an intuitive explanation for the mathematical functionality behind the popularity and success of RNNs in solving practical problems.

## 2 DATA-GENERATION MODEL - A DIFFERENT PERSPECTIVE

Typically, in statistical learning (Shalev-Shwartz & Ben-David, 2014; Vapnik, 1999), it is assumed that the instances of the training data are generated by some probability distribution. For example, we can assume a training input set $\Psi_Y = \{\{\mathbf{y}_i\}_{i=1}^m : \mathbf{y}_i \sim P_Y\}$, such that there is some correct target output $\mathbf{x}$, unknown to the learner, and each pair $(\mathbf{y}_i, \mathbf{x}_i)$ in the training data $\Psi$ is generated by first sampling a point $\mathbf{y}_i$ according to $P_Y(\cdot)$ and then labeling it. The examples in the training set are randomly chosen and, hence, independently and identically distributed (i.i.d.) according to the distribution $P_Y(\cdot)$. We have access to the training error (also referred to as empirical risk), which we normally try to minimize. The known phenomenon of overfitting is when the learning system fits perfectly to the training set and fails to generalize. In classification problems, probably approximately correct (PAC) learning defines the minimal size of a training set required to guarantee a PAC solution. The sample complexity is a function of the accuracy of the labels and a confidence parameter. It also depends on properties of the hypothesis class. To describe generalization we normally differentiate between the empirical risk (training error) and the true risk. It is known that the curse of dimensionality renders learning in high dimension to always amount to extrapolation (out-of-sample extension) (Balestriero et al., 2021).

INFORMATION THEORETIC PERSPECTIVE: CONNECTION TO AEP

Hereafter, we use the notation $x^n$ to denote a sequence $x_1, x_2, ..., x_n$. In information theory, a stationary stochastic process $u^n$ taking values in some finite alphabet $\mathcal{U}$ is called a source. In communication theory we often refer to discrete memoryless sources (DMS) (Kramer et al., 2008; Cover & Thomas, 2006). However, many signals, such as image patches, are usually modeled as entities belonging to some probability distribution forming statistical dependencies (e.g., a Markov-Random-Field (MRF) (Roth & Black, 2009; Weiss & Freeman, 2007)) describing the relations between data points in close spatial or temporal proximity. Here, we will briefly summarize the AEP for ergodic sources with memory (Austin, 2017). Although the formal definition of ergodic process is somewhat involved, the general idea is simple. In an ergodic process, every sequence that is produced by the process is the same in statistical properties (Shannon, 1948). The symbol frequencies obtained from particular sequences will approach a definite statistical limit, as the length of the sequence is increased. More formally, we assume an ergodic source with memory that emits $n$ symbols from a discrete and finite alphabet $\mathcal{U}$, with probability $P_U(u_1, u_2, ..., u_n)$. We recall a theorem (Breiman, 1957), here without proof.

*Theorem* 1 (Entropy and Ergodic Theory). Let $u_1, u_2, ..., u_n$ be a stationary ergodic process ranging over a finite alphabet $\mathcal{U}$, then there is a constant $H$ such that

$$H = \lim_{n \to \infty} -\frac{1}{n} \log_2 P_U(u_1, ..., u_n).$$

$H$ is the entropy rate of the source.

Intuitively, when we observe a source with memory over several time units, the uncertainty grows more slowly as $n$ grows, because once we know the previous sources entries, the dependencies reduce the overall conditional uncertainty. The entropy rate $H$, which represents the average uncertainty per time unit, converges over time. This, of course, makes sense, as it is known that $H(X, Y) \le H(X) + H(Y)$. In other words, the uncertainty of a joint event is less than or equal to the sum of the individual uncertainties. The generalization of the AEP to arbitrary ergodic sources is as following (McMillan, 1953).

*Theorem* 2 (Shannon McMillan (AEP)). For $\epsilon > 0$, the typical set $A_\epsilon^n$ with respect to the ergodic process $P_U(u)$ is the set of sequences $\mathbf{u} = (u_1, u_2, ..., u_n) \in \mathcal{U}^n$ obeying

1. $\Pr[\mathbf{u} \in A_\epsilon^n] > 1 - \epsilon$, for $n$ sufficiently large.

2. $2^{-n(H+\epsilon)} \le P_U(\mathbf{u}) \le 2^{-n(H-\epsilon)}$.

3. $|A_\epsilon^n| \approx 2^{nH}$.

$|A|$ denotes the number of elements in the set $A$, and $\Pr[\mathcal{A}]$ denotes the probability of the event $\mathcal{A}$. In other words, if we draw a random sequence $(u_1, u_2, ..., u_n)$, the typical set has probability nearly 1, all elements of the typical set are nearly equally probable, and the number of elements of the typical set is nearly $2^{nH}$. This property is called the asymptotic equipartition property (AEP). In information theory the AEP is the analog of the law of large numbers (Cover & Thomas, 2006). The notion of a typical sequence was first introduced in 1948 by Shannon in his paper "A Mathematical Theory of Communication" (Shannon, 1948). Intuitively, the typical sequences $u^n$ are the sequences whose *empirical* probability distribution is close to $P_U(\cdot)$.

As mentioned, the entropy rate is more often used for discrete memoryless sources (DMS), yet "every ergodic source has the AEP" (McMillan, 1953). Note that entropy typicality applies also to continuous random variables with a density $p_U$ replacing the discrete probability $P_U(u^n)$ with the density value $p_U(u^n)$. The AEP leads to Shannon's source coding theorem stating that the average number of bits required to specify a symbol in a sequence of length $n$, when we consider only the most probable sequences, is $H$. And it is the foundation for the known rate-distortion theory and channel capacity.

The AEP property divides the space of all possible sequences into two sets: the typical set, where the sample entropy is close to the true entropy, and the non-typical set that consists of the other sequences. We would like to show that most of our attention should be on the typical sequences, because any property that is true for the typical sequences will then be true with high probability and will determine the average behavior of a large sample.

Let us assume a training set $\Psi = \{\mathbf{y}_i, \mathbf{x}_i\}_{i=1}^m$, where $\mathbf{x}_i \in \mathbb{R}^{n \times 1}$ are sampled from $A_\epsilon^n(P_X)$ and paired with $\mathbf{y}_i \in \mathbb{R}^{d \times 1}$ by some function as ground truth. The learning system is trained to output a prediction rule $\mathcal{F} : \mathcal{Y}^d \to \mathcal{X}^n$. Assume an algorithm that trains the predictor by minimizing the training error (empirical error or empirical risk). Assuming a discrete source $P_X(\cdot)$ that emits i.i.d sequences $x^n$ of symbols (for example: patches in an image, segments of

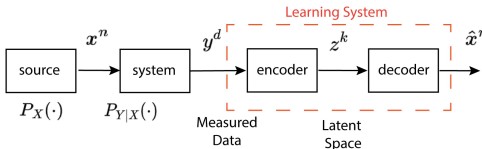

Figure 1: The learning system problem.

an audio signal, etc.), the estimator has access only to the observed signal $y^d$. In an inverse problem $y^d$ would be a degraded signal originating in $x^n$, where the relationship between $x^n$ and $y^d$ could be linear or non-linear, with or without additive noise, such that generally $y^d = g(x^n) + e(x^n)$, where $g(\cdot)$ and $e(\cdot)$ are functions of $x^n$. Given $y^d$, we produce an estimate $\hat{x}^n$. However, the following proofs are not restricted to this framework. Our problem setting is illustrated in Fig. 1. For the sake of the following theoretical analysis we restrict the mapping $\mathcal{F} : \mathcal{Y}^d \to \mathcal{X}^n$ to be a surjective function. Namely, for every $x^n$, there is a $y^d$ such that $f(y^d) = x^n$. In other words, every element $x^n$ is the image of at least one element of $y^d$. It is not required that $y^d$ be unique. In the presence of noise this condition can only be met if the noise's power is under a certain threshold. The goal is to prove that learning with $A_\epsilon^n$ is sufficient for generalization of the entire distribution with the same generalization error. Denote a sample size $|\Psi| = m$ that is required to train a predictor $\mathcal{F}_\Psi : \mathcal{Y}^d \to \mathcal{X}^n$, such that $\Psi \sim P_{X,Y}$. An algorithm minimizes the training error (empirical error or empirical risk)

$$\mathcal{L}_\Psi(\mathcal{F}_\Psi) = \frac{1}{m} \sum_{i=1}^m \ell(\mathcal{F}_\Psi(\mathbf{y}_i), \mathbf{x}_i), \tag{1}$$

where $0 \le \ell(\mathbf{x}, \hat{\mathbf{x}}) \le 1$ is some loss function. The empirical error over the training set at the end of the training, for the specific trained predictor $h_\Psi$ is $\mathcal{L}_\Psi(h_\Psi) \le \Delta_m << 1$. The true error, or the generalization error, in this setting is

$$\mathcal{L}(h_\Psi) = E_{(x,y) \sim P_{X,Y}} \ell(h_\Psi(\mathbf{y}), \mathbf{x}), \tag{2}$$

where $E_{(x,y) \sim P_{X,Y}}(\cdot)$ denotes the expectation over $P_{X,Y}$.

*Theorem* 3 (AEP learning for systems with a surjective mapping). Assume the generalization error of the trained predictor $\mathcal{F} : \mathcal{Y}^d \to \mathcal{X}^n$ over the typical set $A_\epsilon^n(P_X)$ is at most $\varepsilon_A^{\mathcal{F}}$. Then, the generalization error of the trained predictor $\mathcal{F} : \mathcal{Y}^d \to \mathcal{X}^n$ over the entire distribution is at most $\varepsilon_A^{\mathcal{F}}$.

*Theorem* 4 (sample size for learning with a surjective mapping). Assume a training set that is generated by randomly drawing samples from $P_X$ and labeling them by the target function $g(\cdot)$, $\Psi = \left\{ \{\mathbf{y}_i, \mathbf{x}_i\}_{i=1}^m : \mathbf{x}_i \sim P_X, \quad \mathbf{y}_i = g(\mathbf{x}_i), \ \mathbf{x}_i \in \mathbb{R}^{n \times 1}, \ \mathbf{y}_i \in \mathbb{R}^{d \times 1} \right\}$. $g(\cdot)$ is a deterministic function, and $f(\cdot) = g^{-1}(\cdot)$ is a surjective function. Assume the predictor was trained successfully to yield $\mathcal{L}_\Psi(h_\Psi) \leq \Delta_m << 1$. The sample size $m$ and the true error obey

$$\mathcal{L}(h_\Psi) \leq \Delta_m, \quad m \geq 2^{dH(Y)}$$

$$\mathcal{L}(h_\Psi) \leq \Delta, \qquad 2^{dH(Y)} \frac{1 - \Delta}{1 - \Delta_m} \leq m < 2^{dH(Y)}, \quad \Delta_m \leq \Delta \leq 1. \tag{3}$$

In other words, as long as the sample size is larger or equal to the typical set size, we are guaranteed to have a generalization error that is as small as the training error. Otherwise, the upper bound on the generalization error depends on the ratio between the sample size and the input typical set size.

*Theorem* 5 (sample size for learning with a surjective mapping in noisy environment). Assume a training set that is generated by randomly drawing samples from $P_X$ and labeling them by the target function such that, $\Psi = \left\{ \{\mathbf{y}_i, \mathbf{x}_i\}_{i=1}^m : \mathbf{x}_i \sim P_X, \quad \mathbf{y}_i = g(\mathbf{x}_i) + \mathbf{n}_i, \ \mathbf{x}_i \in \mathbb{R}^{n \times 1}, \ \mathbf{y}_i \in \mathbb{R}^{n \times 1} \right\}$, where $g(\cdot)$ is a deterministic known or unknown function, and $\mathbf{n}_i$ is an additive i.i.d noise $\mathbf{n}_i \sim \mathcal{N}^n(\mu, \sigma_n^2)$. The sample size $m$ and the true error obey

$$\mathcal{L}(h_\Psi) \leq \Delta_m, \quad m \geq 2^{nI(X;Y)}$$

$$\mathcal{L}(h_\Psi) \leq \Delta, \quad 2^{nI(X;Y)} \frac{1 - \Delta}{1 - \Delta_m} \leq m < 2^{nI(X;Y)}, \quad \Delta_m \leq \Delta \leq 1, \tag{4}$$

where $I(X;Y) = H(Y) - H(X|Y)$ is the mutual information between $X$ and $Y$.
Note that given $EY^2 < \sigma_x^2 + \sigma_n^2$ we know $I(X;Y) \leq \frac{1}{2} \log(1 + \sigma_x^2/\sigma_n^2)$.

In this case, we can resolve an input $y_i^n$ as $x_i^n$ if they are jointly typical. For each (typical) output sequence $x^n$, there are approximately $2^{nH(Y|X)}$ possible $y^n$ sequences, all of them equally likely. We assume that no two $x^n$ sequences produce the same $y^n$ output sequence, (otherwise, we will not be able to decide which $x^n$ sequence it originated from). Hence, the total number of possible (typical) $y^n$ sequences is approx $2^{nH(Y)}$. This set has to be divided into sets of size $2^{nH(Y|X)}$, corresponding to the different $x^n$ sequences. The total number of disjoint sets is less than or equal to $2^{n(H(Y) - H(Y|X))} = 2^{nI(X;Y)}$. Hence, we can have at most $2^{nI(X;Y)}$ distinguishable sequences of length $n$. The proofs for Theorems 3-5 can be found in Appendix B.

*Remark 1.* Theorem 4 is a specific case of theorem 5, since in our setting, in the absence of noise $I(X;Y) = H(Y)$.
*Remark 2.* The theorems and their proofs can be generalized to the continuous case with $A_\epsilon^n(p_X)$ and differential entropy $h(p_X)$.
*Remark 3.* It is often possible to assume that $\mathcal{X}$ and $\mathcal{Y}$ are discrete alphabets as a result of quantization of real values, and therefore we can discuss discrete entropy.
*Remark 4.* The size of the typical set is exponential by $n$. Therefore, one could claim the size of a typical training set is still large relative to standard training sets that represent the entire probability distribution. However, it is known that data compression and source coding are based on the AEP. And though the size of the typical set is exponential by $n$, compression algorithms, such as JPEG (Wallace, 1991), can compress an image by a factor of 10. In the noiseless case, there are $2^{n \log_2 r}$ possible output sequences, where $r = |\mathcal{Y}|$. Using the AEP it should be possible to train with significantly less examples, because we only try to generalize over $2^{nH(Y)}$ possible output sequences.
*Remark 5*: Zontak & Irani (2011) postulate that patches of the same images are internally repeated, but unlikely to be found in other images. This notion might appear as a contradiction to the proposed analysis. However, note that although the probability of finding a sequence that belongs to $A_\epsilon^n$ is close to one, the probability of finding a specific sequence in the typical set is very small.

The AEP tells us that there exists a relatively small group of training examples that would be sufficient for generalization. However, the AEP property does not define this set, nor the correct coding, learning or prediction method. It just reassures us that there exists a set of the sort. How do we find the typical learning set? One option may be by predefined or learned dictionary coding: build a training set that represents the typical set, consisting of the most common structures, in a similar manner to universal source coding based on a known dictionary (Cover & Thomas, 2006).

## 3 PRELIMINARIES

### 3.1 RNN FRAMEWORK

Assume an observation data sequence $\mathbf{y} = [\mathbf{y}_0, \mathbf{y}_1, ..., \mathbf{y}_{L-1}]$, $\mathbf{y}_t \in \mathbb{R}^{N \times 1}$, $t \in [0, L-1]$, and a corresponding output sequence $\mathbf{x} = [\mathbf{x}_0, \mathbf{x}_1, ..., \mathbf{x}_{L-1}]$, $\mathbf{x}_t \in \mathbb{R}^{P \times 1}$. The RNN forms a map $f : \mathbf{y} \to \mathbf{z}$, from the input data to the latent space variables. That is, for input $\mathbf{y}_t$ and state $\mathbf{z}_t$ of time step $t$, the RNN output is generally formulated as $\mathbf{z}_t = f(\mathbf{z}_{t-1}, \mathbf{y}_t)$ (Pascanu et al., 2013). Hereafter, we will focus on the specific parametrization:

$$\mathbf{z}_t = \sigma(\mathbf{W}_{zy}^T \mathbf{y}_t + \mathbf{W}_{zz}^T \mathbf{z}_{t-1} + \mathbf{b}), \tag{5}$$

where $\sigma$ is an activation function, $\mathbf{W}_{zy} \in \mathbb{R}^{N \times n_{\mathrm{n}}}$ and $\mathbf{W}_{yy} \in \mathbb{R}^{n_{\mathrm{n}} \times n_{\mathrm{n}}}$ are weight matrices and $\mathbf{b} \in \mathbb{R}^{n_{\mathrm{n}} \times 1}$ is the bias vector, assuming $n_{\mathrm{n}}$ number of neurons in an RNN cell. At $t = 0$ previous outputs are zero. Here, we use the ReLU activation function, $\mathrm{ReLU}(z) = \max\{0, z\}$. At this stage, we wrap the cell with a fully connected layer such that the desired final output is $\mathbf{x}_t = \mathrm{FC}(\mathbf{z}_t)$.

Traditionally, RNNs are used for processing of time related signals, to predict future outcomes, and for natural language processing tasks such as handwriting recognition (Graves et al., 2009) and speech recognition (Graves et al., 2013). In computer vision RNNs are less popular, due to gradient exploding and gradient vanishing issues (Pascanu et al., 2013), and their expensive computational complexity compared to CNNs. Liang & Hu (2015) proposed the use of recurrent convolutional networks (RCNNs) for object recognition. Pixel-RNN (Van Den Oord et al., 2016) sequentially predicts pixels in an image along the two spatial dimensions.

### 3.2 SPARSE CODING & ITERATIVE SHRINKAGE ALGORITHMS

In sparse coding (SC) a signal $\mathbf{y} \in \mathbb{R}^{N \times 1}$ is modeled as a sparse superposition of feature vectors (Elad, 2010; Chen et al., 2001). Formally, the observation signal obeys $\mathbf{y} = \mathbf{D}\mathbf{z}$, where $\mathbf{D} \in \mathbb{R}^{N \times M}$ is a dictionary of $M$ atoms $\mathbf{d}_i \in \mathbb{R}^{N \times 1}$, $i = 1, ..., M$, and $\mathbf{z} \in \mathbb{R}^{M \times 1}$ is a *sparse* vector of the atoms weights. Over the years, many efforts have been invested in sparse coding, both in a noise free environment, or when allowing some error,

$$\min_{\mathbf{z}} \|\mathbf{z}\|_1 \qquad \text{s.t.} \qquad \|\mathbf{y} - \mathbf{D}\mathbf{z}\|_2 \leq \varepsilon, \tag{6}$$

where $\|\mathbf{z}\|_1 \triangleq \sum_i |z_i|$, $\|\mathbf{z}\|_2 \triangleq \sqrt{\sum_i z_i^2}$ and $\varepsilon$ is the residual noise or error energy. Further details on sparse coding are in Appendix A.

Consider the cost function,

$$f(\mathbf{z}) = \frac{1}{2} \|\mathbf{y} - \mathbf{D}\mathbf{z}\|_2^2 + \lambda \|\mathbf{z}\|_1, \tag{7}$$

for some scalar $\lambda$. Following Majorization Minimization (MM) strategy, we can build a surrogate function (Daubechies et al., 2004; Elad, 2010)

$$Q(\mathbf{z}, \mathbf{z}_\theta) = f(\mathbf{z}) + d(\mathbf{z}, \mathbf{z}_\theta) = \frac{1}{2} \|\mathbf{y} - \mathbf{D}\mathbf{z}\|_2^2 + \lambda \|\mathbf{z}\|_1 + \frac{c}{2} \|\mathbf{z} - \mathbf{z}_\theta\|_2^2 - \frac{1}{2} \|\mathbf{D}\mathbf{z} - \mathbf{D}\mathbf{z}_\theta\|_2^2. \tag{8}$$

The parameter $c$ is chosen such that the added expression

$$d(\mathbf{z}, \mathbf{z}_\theta) = Q(\mathbf{z}, \mathbf{z}_\theta) - f(\mathbf{z}) = \frac{c}{2} \|\mathbf{z} - \mathbf{z}_\theta\|_2^2 - \frac{1}{2} \|\mathbf{D}\mathbf{z} - \mathbf{D}\mathbf{z}_\theta\|_2^2 \tag{9}$$

is strictly convex, requiring its Hessian to be positive definite, $c\mathbf{I} - \mathbf{D}^T\mathbf{D} \succ \mathbf{0}$. Therefore $c > \|\mathbf{D}^T\mathbf{D}\|_2 = \alpha_{max}(\mathbf{D}^T\mathbf{D})$, i.e., greater than the largest eigenvalue of the coherence matrix $\mathbf{D}^T\mathbf{D}$. The term $d(\mathbf{z}, \mathbf{z}_\theta)$ is a measure of proximity to a previous solution $\mathbf{z}_\theta$. If the vector difference $\mathbf{z} - \mathbf{z}_\theta$ is spanned by $\mathbf{D}$, the distance drops to nearly zero. Alternatively, if $\mathbf{D}$ is not full rank and the change $\mathbf{z} - \mathbf{z}_\theta$ is close to the null space of $\mathbf{D}$, the distance is simply the approximate Euclidean distance between the current solution to the previous one. The sequence of iterative solutions minimizing $Q(\mathbf{z}, \mathbf{z}_\theta)$ instead of $f(\mathbf{z})$, is generated by the recurrent formula $\mathbf{z}_{\theta+1} = \arg\min_{\mathbf{z}} Q(\mathbf{z}, \mathbf{z}_\theta)$, where $\theta \in \mathbb{N}$ is the iteration index. We can find a closed-form solution for its global minimizer that can be intuitively viewed as an iterative projection of the dictionary on the residual term, starting from the initial solution that is a thresholded projection of the dictionary on the observation signal ($\mathbf{z}_0 = \mathbf{0}$):

$$\mathbf{z}_{\theta+1} = \mathcal{S}_{\frac{\lambda}{c}}\left(\frac{1}{c}\mathbf{D}^T(\mathbf{y} - \mathbf{D}\mathbf{z}_\theta) + \mathbf{z}_\theta\right) = \mathcal{S}_{\frac{\lambda}{c}}\left(\frac{1}{c}\mathbf{D}^T\mathbf{y} + (\mathbf{I} - \frac{1}{c}\mathbf{D}^T\mathbf{D})\mathbf{z}_\theta\right), \tag{10}$$

where the $\mathcal{S}_\beta(z) = (|z| - \beta)_+\mathrm{sgn}(z)$ is the soft threshold operator. Assuming the constant $c$ is large enough, it was shown in Daubechies et al. (2004), that (10) is guaranteed to converge to its global minimum. Over time, faster extensions have been suggested, such as: Fast-ISTA (FISTA) (Beck & Teboulle, 2009a), and Learned-ISTA (LISTA) (Gregor & LeCun, 2010),

$$\mathbf{z}_{\theta+1} = \mathcal{S}_{\frac{\lambda}{c}}\left(\mathbf{W}\mathbf{y} + \mathbf{S}\mathbf{z}_\theta\right). \tag{11}$$

$\mathbf{W}$ and $\mathbf{S}$ are learned over a set of training samples $\{\mathbf{y}_i, \mathbf{z}_i\}_{i=1}^m$. $\mathbf{W}$ and $\mathbf{S}$ re-parametrize the matrices $\frac{1}{c}\mathbf{D}^T$ and $\left(\mathbf{I} - \frac{1}{c}\mathbf{D}^T\mathbf{D}\right)$, respectively. Note that $\mathbf{W}$ and $\mathbf{S}$ are no longer constrained by $\mathbf{D}$.

## 4 RNN Analyzed via Sparse Coding

Observing the similar structure of (5) and (11), we redefine the cost function (7)

$$f(\mathbf{z}_t) = \frac{1}{2}\|\mathbf{y}_t - \mathbf{D}\mathbf{z}_t\|_2^2 + \lambda\|\mathbf{z}_t\|_1. \tag{12}$$

Now, building a surrogate function

$$Q(\mathbf{z}_t, \mathbf{z}_{t-1}) = f(\mathbf{z}_t) + d(\mathbf{z}_t, \mathbf{z}_{t-1}) = \frac{1}{2}\|\mathbf{y}_t - \mathbf{D}\mathbf{z}_t\|_2^2 + \lambda\|\mathbf{z}_t\|_1 + \frac{c}{2}\|\mathbf{z}_t - \mathbf{z}_{t-1}\|_2^2 - \frac{1}{2}\|\mathbf{D}\mathbf{z}_t - \mathbf{D}\mathbf{z}_{t-1}\|_2^2, \tag{13}$$

where the added term $d(\mathbf{z}_t, \mathbf{z}_{t-1})$ represents the distance between the *current solution* and the *previous solution* at the *preceding time step* (rather than the previous iteration), yields

$$\mathbf{z}_t = \mathcal{S}_{\frac{\lambda}{c}}\left(\frac{1}{c}\mathbf{D}^T(\mathbf{y}_t - \mathbf{D}\mathbf{z}_{t-1}) + \mathbf{z}_{t-1}\right), \tag{14}$$

which in its learned version can be re-parametrized as,

$$\mathbf{z}_t = \mathcal{S}_\beta\left(\mathbf{W}_{zy}^T\mathbf{y}_t + \mathbf{W}_{zz}^T\mathbf{z}_{t-1}\right). \tag{15}$$

Clearly, (15) is equivalent to (5). In other words, *a RNN can be viewed as unfolding of one iteration of a learned sparse coder, based on an assumption that the solution at time $t$ is close to the solution in time $t - 1$. In subsection 3.1, the RNN state encodes the vector to a latent space. Then the FC net decodes the latent variable back to a space of the required dimensions. Given this interpretation, it may be claimed that RNN's use should not be restricted to data with obvious time or depth relations. The RNN merely serves as an encoder providing a rough estimation of the sparse code of the input data. The RNN's memory serves the optimization process by starting the computation from a closer solution. Thus placing the initial solution in a "close neighborhood" or close proximity, and helping the optimization gravitate more easily towards the latent space sparse approximation. Clearly, convergence is not guaranteed.

## 5 Few Shot Learning via RNN

The setting described in this subsection has been previously employed for various applications. Biswas et al. (2018) and Pereg et al. (2020a) used a similar framework to facilitate automatic velocity analysis, where they used a portion of the acquired data for training and let the system infer the rest of the missing velocities. Pereg et al. (2020b) used a similar framework to perform seismic inversion with synthetic training data. Here, we expand and elaborate its application, while connecting it to the theoretical intuition in Section 2. The description below is formulated for two-dimensional (2D) input signals, but can be easily adapted to other input data dimensions. We use similar definitions and notations as previously described in (Pereg et al., 2020a;b).

*Definition 1 (Analysis Patch):* We define an *analysis patch* as a 2D patch of size $L_t \times N$ enclosing $L_t$ time (depth) samples of $N$ consecutive neighboring columns of the observed image $\mathbf{S} \in \mathbb{R}^{L_r \times J}$. Assume $\{n_L, n_R \in \mathbb{N} : n_L + n_R = N - 1\}$. The analysis patch $\mathbf{A}^{(i,j)}$ associated with an image point at location $(i, j)$, such that element $(k, l)$ of $\mathbf{A}^{(i,j)}$ is

$$\mathbf{A}_{k,l}^{(i,j)} = \{S_{i+k,j+l} : k, l \in \mathbb{Z}, 1 - L_t \le k \le 0, -n_L \le l \le n_R\}.$$

An analysis patch $\mathbf{A}^{(i,j)} \in \mathbb{R}^{L_t \times N}$ is associated with a pixel $\mathbf{R}_{i,j}$ in the output image. To produce a point in the estimated $\hat{\mathbf{R}}_{i,j}$, we set an input to the RNN as $\mathbf{y} = \mathbf{A}^{(i,j)}$. Each time step input is a group of $N$ neighboring pixels of the same corresponding time (depth). In other words, in our application $\mathbf{y}_t = [S_{i-L_t+1+t,j-n_L}, ..., S_{i-L_t+1+t,j+n_R}]^T$. We set the size of the output vector $\mathbf{x}_t$ to one expected pixel ($P = 1$), such that $\mathbf{x}$ is expected to be the corresponding segment, $\mathbf{x} = [R_{i-(L_t-1),j}, ..., R_{i,j}]^T$. Lastly, we ignore the first $L_t-1$ values of the output $\mathbf{x}$ and set the predicted pixel $\hat{R}_{i,j}$ as the last one, $x_{L_t}$. The analysis patch moves through the input image and produces all expected output points in the same manner. Each analysis patch and a corresponding output vector are an instance for the net. The size and shape of the analysis patch defines the geometrical distribution of columns and samples that are considered in each output point computation.

Note that, in the above framework the number of hidden units is typically much larger than the dimension of the input, $n_n >> N$, ($N << M$). The dimension of the latent space can be 3 orders of magnitude larger than the input space dimension. In accordance with that, it has been observed that over-parametrized neural networks generalize better (Neyshabur et al., 2018) and that dictionary recovery may be facilitated with over-realized models (Sulam et al., 2022).

**RFN-RNN.** Employing one projection assuming $\mathbf{x}_{t-1}$ is close to $\mathbf{x}_t$ may throw us far away from the desired solution. RFN (Pereg et al., 2021) refers to local signal normalization, in an attempt to find the detected features' support in fewer iterations, by avoiding the liability of unbalanced energy. Since the RNN is essentially a sparse solver, we propose to apply 2D-RFN, prior to applying the RNN. We then multiply the latent space support normalized weights by the non-normalized projection of the learned dictionary on the original input signal to regain the local energy. Appendix C describes 2D-RFN, and provides the proposed method's forward pass in Algorithm 1. The RFN signal's distribution is confined to a smaller set of values. Thus, RFN decreases the signal's entropy and the size of the typical set associated with the input distribution.

## 6 NUMERICAL EXPERIMENTS

### 6.1 CASE STUDY: IMAGE DEBLURRING

We studied the RNN framework in Section 5 and Algorithm 1 (RNN-RFN) for Gaussian deblurring. First, we follow the same Gaussian deblurring experiment performed in Romano et al. (2017), using the set of natural images provided by the authors. All pixel values are in the range of [0,255]. The images were convolved with a 2D Gaussian PSF $25 \times 25$ with standard deviation of 1.6. RGB images are converted to YCbCr color-space. Inversion is applied to the luminance channel, and the result is converted back to the RGB domain. As a figure of merit, we used the peak signal to noise ratio (PSNR) and the SSIM, both computed on the estimated luminance channel of the ground truth and the estimated image. We investigated the basic RNN and RNN-RFN frameworks, and studied the effect of different training and testing images, with additive WGN with $\sigma_n = \sqrt{2}$.

For each experiment, we used one of the images for training and the remaining 9 images for testing. The assumption that the latent space representation is sparse was verified in the numerical experiments. Table 1 presents the optimal PSNR and SSIM scores obtained for each image, and compares the PSNR scores to state-of-the-art image deblurring methods: FISTA (Beck & Teboulle, 2009b), NCSR (Dong et al., 2013) and RED (Romano et al., 2017). Note that most deblurring methods require prior knowledge of the degradation process. Whereas the proposed approach requires only one example of the degraded image and its corresponding ground truth. The best scores were obtained by training with either one of the images: butterfly, boats, parrot, starfish and peppers. Hence, it is safe to assume that the patches in these images reliably represent the typical set associated with natural images to a certain degree. Figures 2-3 present examples of the proposed methods. Figure 6 show the evolution of the PSNR scores for RNN and RNN-RFN with increasing additive noise variance $\sigma_n \in [0, 10\sqrt{2}]$. As the noise level is increased RFN suppresses noise better. Additional examples and training details are in Appendix D.1.

At this stage, we are not suggesting that the results are competitive with state-of-the-art denoisers. Currently, the main advantage is substantial speed. Tensorflow (Abadi et al., 2016) implementation converges in about 1700 iterations in an average of **14.29 seconds** on a laptop GPU (NVIDIA GeForce GTX Ti 1650 with 4GB video memory) and 2.01 minutes on i-7 CPU. Pytorch

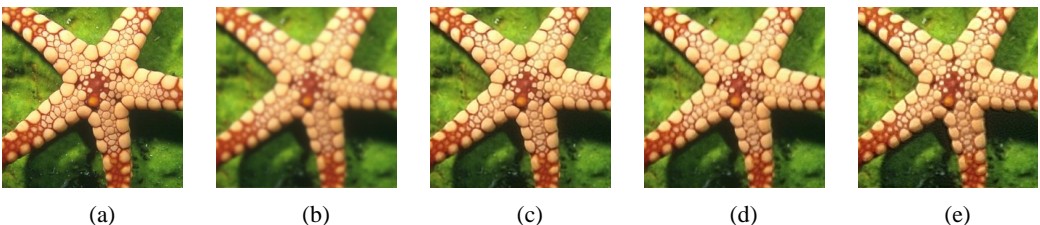

|     |     |     |     |     |
| (a) | (b) | (c) | (d) | (e) |

Figure 2: Visual comparison of deblurring of the image starfish: (a) Ground truth; (b) input, 24.8dB; (c) RED-SD, PSNR=32.42dB; (d) RNN 29.18dB; (e) RNN-RFN, 28.45dB.

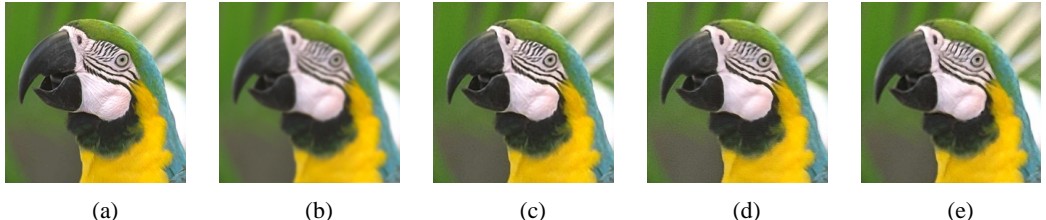

|     |     |     |     |     |
| (a) | (b) | (c) | (d) | (e) |

Figure 3: Visual comparison of deblurring of the parrot image: (a) Ground truth; (b) input, 25.33dB; (c) RED-SD, PSNR=33.18dB; (d) RNN 30.10dB; (e) RNN-RFN, 29.64dB;

(Paszke et al., 2019) implementation training converges in 40 epochs on average in 30.73 seconds on a GPU NVIDIA GeForce RTX 2080 Ti, and 2.73 minutes on i-7 CPU. Inference takes ∼500 msec. For reference, RED-SD takes 15-20 minutes of processing for each image, on a CPU.

As the focus of this study is to bridge between theory and application for few-shot learning, and to provide theoretical justifications, rather than to achieve state-of-the-art results, we verified the theoretical results in Theorem 5 for the image deblurring case. To this end, we trained the RNN network with varying sample sizes while trying to keep a constant training error, $\mathcal{L}_\Psi(h_\Psi) \leq \Delta_m$. The training data consists of patches belonging to a single image (starfish). We consider the recovery error, denoted as $\hat{\mathcal{L}}(h_\Psi)$, over the other non-training 9 images as an empirical approximation for the generalization error. Figure 9 shows the evolution of the recovery error with increasing sample size chosen uniformly in the range [1, 61752]. Evidently, $\hat{\mathcal{L}}(h_\Psi) \geq \mathcal{L}_\Psi(h_\Psi)$, while $\mathcal{L}_\Psi(h_\Psi) \leq \Delta_m$, which implies that $m < 2^{I(X;Y)}$. In this region $\mathcal{L}(h_\Psi) \leq 1 - m2^{-nI(X;Y)}(1 - \Delta_m) \leq \Delta$ (see proof). Hence, empirically we can deduce $I(X;Y) \geq 0.33$. Our code will be available upon acceptance.

## 6.2 CASE STUDY: OCT SPECKLE SUPPRESSION

OCT uses low coherence interferometry to produce cross-sectional tomographic images of internal structure of biological tissue. It is routinely used for diagnostic imaging of the anterior eye, the retina, and, through a fiber-optic catheter, the coronary arteries. Unfortunately, OCT images are degraded by the presence of speckle (Schmitt et al., 1999; Goodman, 2007), which appear as grain-like structures in the image, with a size equivalent to the nominal spatial resolution of the OCT system. Speckle significantly degrade images and complicate interpretation and medical diagnosis by confounding tissue anatomy and masking changes in tissue scattering properties. Though speckle share many statistical properties with noise, it is essentially unresolved spatial information as a result of the interference of many sub-resolution spaced scatterers (Curatolo et al., 2013).

Here, we show two examples of supervised learning speckle suppression with OCT experimental data, for demonstration. We used intensity OCT tomograms displaying the log-scaled squared norm of the complex-valued tomogram. As ground truth for training and for evaluation of the generalization, we used hardware-based speckle mitigation obtained by dense angular compounding, in a method similar to Desjardins et al. (2007). We investigated two challenging cases of mismatch between the training data and the testing data: (1) Tissue type mismatch. (2) Tissue type and acquisition system mismatch. Namely, in the first case, OCT tomograms of chicken tissue and a blueberry were acquired by the same system and used as training image and testing image, respectively. Whereas, in the second case, the training image was a tomogram of chicken-skin acquired by one system, and the learning performance was tested on a tomogram of cucumber imaged with a different OCT system. Further details are in Appendix D.2.

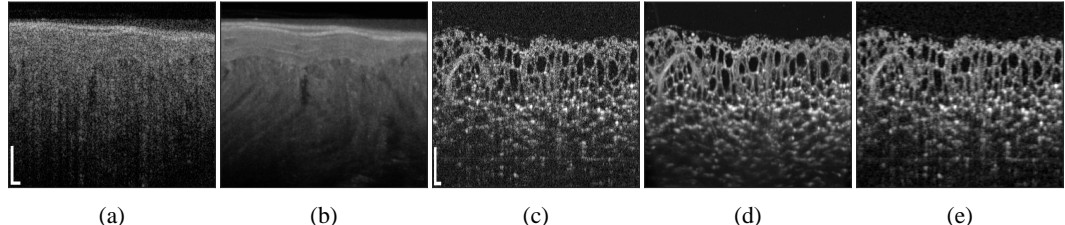

(a)          (b)          (c)          (d)          (e)

Figure 4: OCT speckle suppression with tissue mismatch and identical imaging system: Training images: (a) Chicken muscle with speckle (PSNR = 26.03 dB); (b) Ground truth, speckle-suppressed tomogram; (c) Blueberry with (PSNR = 21.39 dB) and (d) without speckle (ground truth); (e) Predicted speckle-suppressed tomogram (PSNR=31.1dB). Scale bars: $200\mu$m.

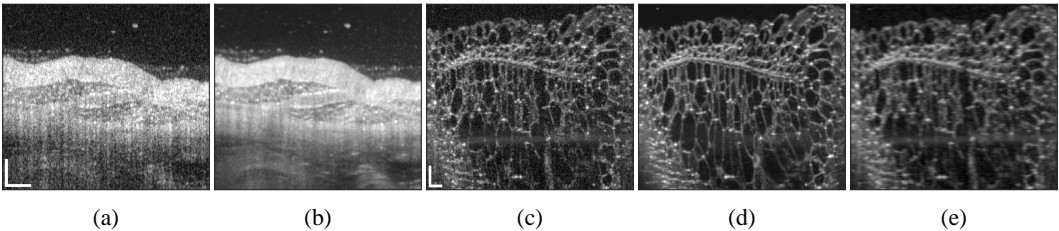

(a)          (b)          (c)          (d)          (e)

Figure 5: OCT speckle suppression with tissue and system mismatch: Training images: (a) Chicken skin with speckle (PSNR = 28.96 dB); (b) Ground truth, speckle-suppressed tomogram; (c) Cucumber with (PSNR = 29.40 dB) and (d) without speckle (ground truth), imaged with inferior lateral resolution and lower lateral sampling than training image; (e) Predicted speckle-suppressed tomogram (PSNR=31.34dB). Scale bars: $200\mu$m.

Figures 4(a)-(b) show the speckle-corrupted and speckle-suppressed tomograms of chicken tissue used for training. The speckle-suppressed tomogram in Fig. 4(b) has a homogeneous appearance, whereas the speckled signal in Fig. 4(a) is strongly textured, exhibiting a grainy pattern with spots of bright, high intensity signal, alternating with regions of very low signal. Figures 4(c)-(d) show the speckle-corrupted and speckle-suppressed tomograms of the blueberry used for testing. As shown in the inferred result in Fig. 4(e), the system is able to significantly reduce speckle while maintaining spatial resolution, despite the structural differences between the training and the testing images.

Figures 5(a)-(b) show the original and compounded tomograms of chicken skin used for training. Visibly the training data is non-stationary and of complex structure. Figures 5(c)-(d) show the original and compounded tomograms of cucumber used for testing. The cucumber image was acquired with a different sample objective lens resulting in twice poorer lateral resolution. The lateral sampling space was increased from $2.5\mu$m to $8\mu$m, resulting in system mismatch between training and testing data, in addition to the tissue mismatch. As can be seen, the learning system is able to detect most features and to significantly remove speckle. Horizontal lines appear slightly sharper than vertical ones.

**Computational resources.** The proposed framework training time is only 35.00 seconds on a standard CPU workstation equipped with an Intel(R) Core(TM) i7 -7820HQ CPU @ 2.30 GHz, or alternatively **4.56 seconds** on average on a laptop GPU NVIDIA GeForce GTX Ti 1650 with 4GB video memory, and converges typically in less than 700 iterations.

## 7   CONCLUSIONS

"The curse of dimensionality can be viewed either as the limitation on data analysis due to the large amount of data or parameters needed to analyze the data" (Bengio et al., 2007). In this work, we investigated the properties of the AEP in the context of supervised learning, and its practical ramifications in relieving computational complexity in learning systems. We focused on a RNN framework as a powerful tool for learning with limited training data. The low computational complexity can be leveraged for medical areas where there is lack of data and resources. Future work could extend the proposed work to other tasks and applications in other domains. Our future work will extend the AEP hypothesis to the automatic design of the AEP training dataset for unsupervised medical imaging learning tasks.

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

## A  SPARSE REPRESENTATIONS

Sparse coding (SC) is a popular task in many fields, such as: image processing (Elad, 2010), computer vision (Jarrett et al., 2009), compressed sensing (Donoho, 2006), ultrasound imaging (Bendory et al., 2016), seismology (Pereg & Cohen, 2017; Pereg et al., 2017; 2019; 2021), and visual neurosciense (Olshausen & Field, 1996; Lee et al., 2009).

A sparse representations model (Elad, 2010) assumes a signal $\mathbf{y} \in \mathbb{R}^{d \times 1}$ that is analyzed as a sparse linear combination of some dictionary basis components:

$$\mathbf{y} = \mathbf{Dz}, \tag{16}$$

where $\mathbf{D} \in \mathbb{R}^{d \times M}$ is a matrix called the dictionary, built of the atoms $\mathbf{d}_i \in \mathbb{R}^{d \times 1}$, $i = 1, ..., M$, as its columns, and $\mathbf{z} \in \mathbb{R}^{M \times 1}$ is the sparse vector of the atoms weights. Sparse coding, that is, the recovery of $\mathbf{z}$, has been the center of tremendous research efforts. Finding the sparsest solution, the one with the smallest $\ell_0$-norm, is basically attempting to solve

$$(P_0): \qquad \min_{\mathbf{z}} \|\mathbf{z}\|_0 \qquad \text{s.t.} \qquad \mathbf{y} = \mathbf{Dz}, \tag{17}$$

where $\|\mathbf{z}\|_0$ denotes the number of non-zeros in $\mathbf{z}$. Unfortunately, $P_0$ is in general NP-Hard (Natarajan, 1995), therefore the $\ell_0$-norm is often replaced with the $\ell_1$-norm

$$(P_1): \qquad \min_{\mathbf{z}} \|\mathbf{z}\|_1 \qquad \text{s.t.} \qquad \mathbf{y} = \mathbf{Dz}, \tag{18}$$

where $\|\mathbf{z}\|_1 \triangleq \sum_i |z_i|$. In many real-life scenarios, such as in the presence of noise or when some error is allowed, we solve

$$(P_{1,\varepsilon}): \qquad \min_{\mathbf{z}} \|\mathbf{z}\|_1 \qquad \text{s.t.} \qquad \|\mathbf{y} - \mathbf{Dz}\|_2 \leq \varepsilon, \tag{19}$$

where $\|\mathbf{z}\|_2 \triangleq \sqrt{\sum_i z_i^2}$. The sparsest solution to $P_0$ and $P_1$ is unique under certain conditions, and can be obtained with known algorithms, such as orthonormal matching pursuit (OMP) or basis pursuit (BP), depending on the dictionary's properties and the degree of sparsity of $\mathbf{z}$. That is, when $\|\mathbf{z}\|_0 < \frac{1}{2}\left(1 + \frac{1}{\mu(\mathbf{D})}\right)$, where $\mu(\mathbf{D})$ is the mutual coherence defined as

$$\mu(\mathbf{D}) = \max_{i \neq j} \frac{\left|\mathbf{d}_i^T \mathbf{d}_j\right|}{\|\mathbf{d}_i\|_2 \cdot \|\mathbf{d}_j\|_2}, \tag{20}$$

the true sparse code $\mathbf{z}$ can be perfectly recovered (Candès et al., 2011).

An intuitive way to recover $\mathbf{z}$ is to project $\mathbf{y}$ on the dictionary, and then extract the atoms with the strongest response by taking a hard or a soft threshold, i.e., $\mathbf{z} = \mathcal{H}_\beta(\mathbf{D}^T \mathbf{y})$ or $\mathbf{z} = \mathcal{S}_\beta(\mathbf{D}^T \mathbf{y})$, where the hard threshold and the soft threshold operators are respectively defined as

$$\mathcal{H}_\beta(z) = \begin{cases} z, & |z| > \beta \\ 0, & |z| \leq \beta \end{cases}, \quad \text{and} \quad \mathcal{S}_\beta(z) = \begin{cases} z + \beta, & z < -\beta \\ 0, & |z| \leq \beta \\ z - \beta, & z > \beta \end{cases}.$$

Note that the ReLU activation function obeys

$$\mathrm{ReLU}(z - \beta) = \max(z - \beta, 0) = \mathcal{S}_\beta^+(z) \triangleq \begin{cases} 0, & z \leq \beta \\ z - \beta, & z > \beta \end{cases}.$$

Therefore, the soft threshold solution can be also written as

$$\mathbf{z} = \mathcal{S}_\beta^+(\mathbf{D}^T \mathbf{y}) - \mathcal{S}_\beta^+(-\mathbf{D}^T \mathbf{y}) = \mathrm{ReLU}(\mathbf{D}^T \mathbf{y} - \beta) - \mathrm{ReLU}(-\mathbf{D}^T \mathbf{y} - \beta).$$

It is possible to assume a nonnegative sparse code such that the weights are solely positive, essentially assuming a compounded dictionary $[\mathbf{D}, -\mathbf{D}]$ (Papyan et al., 2017). Hence a nonnegative model does not affect the expressiveness of the model. Perfect support recovery by simple thresholding is guaranteed only when $\|\mathbf{z}\|_0 < \frac{1}{2}\left(1 + \frac{1}{\mu(\mathbf{D})}\frac{|\mathbf{z}|_{\min}}{|\mathbf{z}|_{\max}}\right)$, where $|\mathbf{z}|_{\min}$, and $|\mathbf{z}|_{\max}$ are the minimum and maximum values of the vector $|\mathbf{z}|$ on the support, implying that this approach may have stability issues when the data is unbalanced.

In the special case where $\mathbf{D}$ is a convolutional dictionary, the task of extracting $\mathbf{z}$ is referred to as convolutional sparse coding (CSC). In this case, the dictionary $\mathbf{D}$ is a convolutional matrix constructed by shifting a local matrix of filters in all possible positions. Papyan et al. (2017) show that the forward pass of CNNs is equivalent to the layered thresholding algorithm designed to solve the CSC problem.

## B  PROOFS

### B.1  PROOF OF THEOREM 3

We assume a training set $\Psi$, where $\{\mathbf{x}_i\}_{i=1}^m$ are sampled from $A_\epsilon^n(P_X)$ and labeled by some function as ground truth, $\Psi = \{\{\mathbf{y}_i, \mathbf{x}_i\}_{i=1}^m : \mathbf{x}_i \in A_\epsilon^n(P_X)\}$. The output of a predictor $\mathcal{F} : \mathcal{Y} \to \mathcal{X}$. Assume an algorithm that minimizes the training error (empirical error or empirical risk)

$$\mathcal{L}_\Psi(\mathcal{F}_\Psi) = \frac{1}{m}\sum_{i=1}^m \ell(\mathcal{F}_\Psi(\mathbf{x}_i), \mathbf{y}_i) \tag{21}$$

Assuming a discrete source $P_X(\cdot)$ that emits i.i.d sequences $x^n$ of symbols (for example: patches in an image, segments of an audio signal, etc.). The estimator has access only to the observed signal $y^d$. (In an inverse problem $y^d$ would be a degraded signal originating in $x^n$, where the relationship between $x^n$ and $y^d$ could be linear or non-linear, deterministic or stochastic. Generally, $y^d = g(x^n) + e(x^n)$.) Given $y^d$, we produce an estimate $\hat{x}^n$. However the following proof is not restricted to this framework. For the sake of the following theoretical analysis we restrict the mapping $\mathcal{F} : \mathcal{Y}^d \to \mathcal{X}^n$ to be a surjective function. In other words, for every $x^n$, there is a $y^d$ such that $f(y^d) = x^n$. In other words, every element $x^n$ is the image of at least one element of $y^d$. It is not required that $y^d$ be unique. In the presence of noise this condition can only be met if the noise's power is under a certain threshold. The goal is to prove that learning with $A_\epsilon^n(P_X)$ is equivalent to training with the entire distribution with the same generalization error.

A sequence $y^d$ of symbols is passed to the learning system. For example, it is possible that this system is built as an encoder-decoder, such that the encoder "compresses" $y^d$ into a latent space representation vector $z^k$ and sends $z^k$ into the decoder. The decoder reconstructs $x^n$ from $z^k$, as $\hat{x}^n(z^k)$. Generally speaking, the predictor reconstruct $\hat{x}^n$ from $y^d$ and is said to be successful if $\hat{x}^n = x^n$. We consider the case where every source sequence $x^n$ is assigned a unique $z^k$. Therefore one can reconstruct $x^n$ perfectly. Note that the same latent space $z^k$ can represent different observed sequences $y^d$. This assumption is true if and only if the mapping $\mathcal{F} : \mathcal{Y}^d \to \mathcal{X}^n$ is unique. In other words, every $x^n$ can be mapped to more than one $y^d$, but every $y^d$ can only be mapped to one $x^n$. The goal is to prove that learning by training over the typical set $A_\epsilon^n(P_X)$ is sufficient.

Denote $A = \{x^n : x^n \in A_\epsilon^n(P_X)\}$, $B = \{x^n : x^n \notin A_\epsilon^n(P_X)\}$, $|B| = \delta_\epsilon(n)$. We train the predictor with a training set $\Psi_A = \{\{y_i^d, x_i^n\}_{i=1}^m : x_i^n \in A\}$, such that the generalization error (risk) over $x^n \in A$ is at most $\varepsilon_A^\mathcal{F}$. Now, given some test input $y^d$, if $x^n(y^d) \in A_\epsilon^n(P_X)$ the encoder passes to the decoder the $z^k$ that represents this sequence. In the general case, the predictor deciphers $y^d$ as trained by generalization. However if $x^n(y^d) \notin A_\epsilon^n(P_X)$ we can assume some unknown output $\hat{x}^n$ (the encoder sends to the decoder some unknown $z^k$ generated by the trained learning system), with error $\varepsilon_B^\mathcal{F}$. The average error is upperbounded by

$$\mathcal{L}(h_\Psi) \le Pr\{x^n \in A_\epsilon^n(P_X)\}\varepsilon_A^\mathcal{F} + Pr\{x^n \notin A_\epsilon^n(P_X)\}\varepsilon_B^\mathcal{F} \le \varepsilon_A^\mathcal{F} + \delta_\epsilon(n)\varepsilon_B^\mathcal{F}. \tag{22}$$

But since $\delta_\epsilon(n) \to 0$ as $n \to \infty$, we have $\mathcal{L}(h_\Psi) \le \varepsilon_A^\mathcal{F}$. $\qquad\square$

An alternative way to derive the same result is as following. The generalization error of the trained predictor $h_\Psi$ is

$$\mathcal{L}(h_\Psi) = E_{(x,y) \sim P_{X,Y}} \, \ell\big(h_\Psi(y), x\big) = \sum_{(x,y) \sim P_{X,Y}} P_{X,Y}(x,y) \, \ell\big(h_\Psi(y), x\big)$$

$$= \sum_{x \sim P_X} P_X(x) \sum_{y \sim P_{Y|X}} P_{Y|X}(y|x) \, \ell\big(h_\Psi(y), x\big)$$

$$= \sum_{x \in A_\epsilon^n(P_X)} P_X(x) \sum_{y \sim P_{Y|X}} P_{Y|X}(y|x) \, \ell\big(h_\Psi(y), x\big) = \mathcal{L}_A(h_\Psi) \leq \varepsilon_A^{\mathcal{F}},$$

where $\mathcal{L}_A(h_\Psi)$ denotes the generalization error over the typical set $A_\epsilon^n(P_X)$. The third equality follows from $Pr[x^n \in A_\epsilon^n(P_X)] = 1$. Throughout the proofs we sometimes omit the superscript $^n$ for simplicity. Note that this derivation is symmetric for $x$ and $y$, therefore it is possible to build a training set by drawing samples $y^d \in A_\epsilon^d(P_Y)$ and pairing them with the corresponding $x^n$, under the assumption that the mapping $\mathcal{F} : \mathcal{Y}^d \to \mathcal{X}^n$ is a surjective function. Alternatively, it is possible to define the jointly typical set $\mathcal{B} = \{(x^n, y^d) : (x^n, y^d) \in A_\epsilon^{n,d}(P_{X,Y}), y^n \in A_\epsilon^d(P_Y), x^n \in A_\epsilon^n(P_X)\}$ and assume the training input-output pairs are drawn from $\mathcal{B}$.

## B.2   PROOF OF THEOREM 4

Assume a supervised learning algorithm with a training set $\Psi$, sampled from an unknown distribution $P_X$ and paired with $y$ by some target function $g(\cdot)$, to learn a predictor $h_\Psi : \mathcal{Y}^d \to \mathcal{X}^n$. (Here, the subscript $\Psi$ emphasizes that the output predictor depends on $\Psi$.) The goal of the algorithm is to find $\mathcal{F}_\Psi$ that minimizes the error with respect to the unknown $P_{X,Y}(\cdot)$ over $\mathcal{X}^n \times \mathcal{Y}^d$. Since the learner does not know $P_{X,Y}(\cdot)$, the true error is not available to the learner. A useful notion of error that can be calculated by the learner is the training error (the error over the training sample), a.k.a the empirical error or empirical risk,

$$\mathcal{L}_\Psi(h) = \frac{1}{m} \sum_{i=1}^{m} \ell\big(h(\mathbf{x}_i), \mathbf{y}_i\big) \tag{23}$$

where $m = |\Psi|$ and $\ell(x, \hat{x})$ is some loss function, such as: MSE, $\ell_1$ norm, etc. The training sample is only a glimpse into the world that is available to the learner. This learning paradigm coming up with a predictor that minimizes $\mathcal{L}_\Psi(\mathcal{F}_\Psi)$ is called Empirical Risk Minimization or ERM for short (Shalev-Shwartz & Ben-David, 2014). We define the true error, or the generalization error, in our framework as

$$\mathcal{L}(h_\Psi) = E_{(x,y) \sim P_{X,Y}} \ell\big(h_\Psi(y), x\big), \tag{24}$$

where $E_{(x,y) \sim P_{X,Y}}(\cdot)$ denotes the expectation over $P_{X,Y}$.

Let the learner's output sequence $\mathbf{x} \in \mathbb{R}^{n \times 1}$ be a finite sequence whose i'th entry $x_i$ takes on values in a discrete and finite alphabet $\mathcal{X}$. We write $\mathcal{X}^n$ for the Cartesian product of the set $\mathcal{X}$ with itself $n$ times. In other words, our hypothesis class $\mathcal{H}$ obeys $\mathcal{H} = \{h : \mathcal{Y}^d \to \mathcal{X}^n\}$. For a training sample $\Psi$ the trained predictor $h_\Psi$, minimizing (23),

$$h_\Psi \in \arg\min_{h \in \mathcal{H}} \mathcal{L}_\Psi(h)$$

Assume the predictor was trained successfully to yield $\mathcal{L}_\Psi(h_\Psi) \leq \Delta_m \ll 1$. Recall the sequences that serve as the examples in the training set are independently and identically distributed (i.i.d.). We are interested in finding the sample size of $m$ sequences of instances that will lead to a bounded generalization error for the specific trained predictor, $\mathcal{L}(h_\Psi) \leq \Delta$. Namely,

$$\mathcal{L}(h_\Psi) = E_{(x,y) \sim P_{X,Y}} \, \ell\big(h_\Psi(y), x\big) = \sum_{x,y} P_{X,Y}(x,y) \, \ell\big(h_\Psi(y), x\big)$$

$$= \sum_{y} P_Y(y) \, \ell\big(h_\Psi(y), f(y)\big) = \sum_{y \in A_\epsilon^d(P_Y)} P_Y(y) \, \ell\big(h_\Psi(y), f(y)\big),$$

the second equation follows from $P(X,Y) = P(Y)P(X|Y)$. Recall that we assumed $g(\cdot)$ is a deterministic function, and that $f(\cdot)$ is a subjective function, therefore $P(X = f(y)|Y = y) = 1$.

And the third equation follows from $Pr[y^d \in A_\epsilon^d] = 1$. If $m \geq 2^{dH(Y)}$, and since $|A_\epsilon^d(P_Y)| = 2^{dH(Y)}$, and $P_Y(y^d) = 2^{-dH(Y)}$, then clearly $\mathcal{L}(h_\Psi) = \mathcal{L}_\Psi(h_\Psi) \leq \Delta_m$. Otherwise, $m < 2^{dH(Y)}$ and

$$\mathcal{L}(h_\Psi) \leq (2^{dH(Y)} - m)2^{-dH(Y)} + m\Delta_m 2^{-dH(Y)} \leq \Delta,$$

where we have used the fact that $l(x, \hat{x}) \leq 1$. We now have

$$1 - m2^{-dH(Y)}(1 - \Delta_m) \leq \Delta, \tag{25}$$

And since $\Delta_m << 1$,

$$m \geq 2^{dH(Y)}\frac{1 - \Delta}{1 - \Delta_m} \approx 2^{dH(Y)}(1 - \Delta)(1 + \Delta_m), \tag{26}$$

or approximately

$$m \geq 2^{dH(Y)}(1 - \Delta). \tag{27}$$

Equivalently,

$$\Delta \geq 1 - \frac{m}{2^{dH(Y)}}(1 - \Delta_m). \tag{28}$$

Hence the ratio between the sample size and the input data typical set determines the upper bound on the generalization error. $\qquad\square$

### B.3 PROOF OF THEOREM 5

Since $y = g(x) + n$, each of the possible $y$ sequences induces a probability distribution on the $x$ sequences. Since two different $y^n$ may originate in the same $x^n$ sequence, the $x^n$ are confusable. For a "non-confusable" subset of input sequences, such that with high probability there is only one highly likely $x^n$ that could have caused the particular output, we can reconstruct the sequence at the output with a negligible probability of error, by mapping the observations into the appropriate "widely spaced" hidden sequences. We can define the conditional entropy $H(Y|X)$ assuming they are ergodic and have a stationary coupling (Gray, 2011). Defining their mutual information $I(X;Y) = H(Y) - H(Y|X)$, their jointly typical set follows similar properties (Cover & Thomas, 2006). Define $B = \{(x^n, y^n) : (x^n, y^n) \in A_\epsilon^n(P_{X,Y}), y^n \in A_\epsilon^n(P_Y), x^n \in A_\epsilon^n(P_X)\}$, and $A_\epsilon^n(P_{X,Y}|x^n) = \{y^n : (x^n, y^n) \in A_\epsilon^n(P_{X,Y})\}$. Observe that $A_\epsilon^n(P_{X,Y}|x^n) = \emptyset$ if $x^n \notin A_\epsilon^n(P_X)$ (Kramer et al., 2008).

$$\Pr[Y^n \in A_\epsilon^n(P_{X,Y}|x^n)|X^n = x^n] \approx 1. \tag{29}$$

$$\Pr[(x^n, y^n) \in B] \approx 2^{-nI(X;Y)}. \tag{30}$$

$$|B| \approx 2^{nI(X;Y)}. \tag{31}$$

Roughly speaking, we can resolve an input $y_i^n$ as $x_i^n$ if they are jointly typical. For each (typical) output $x^n$ sequence, there are approximately $2^{nH(Y|X)}$ possible $y^n$ sequences, all of them equally likely. We wish to ensure that no two $x^n$ sequences produce the same $y^n$ output sequence, otherwise, we will not be able to decide which $x^n$ sequence it was originated from. The total number of possible (typical) $y^n$ sequences is approx $2^{nH(Y)}$. This set has to be divided into sets of size $2^{nH(Y|X)}$ corresponding to the different input X sequences. The total number of disjoint sets is less than or equal to $2^{n(H(Y)-H(Y|X))} = 2^{nI(X;Y)}$. Hence, we can have at most $2^{nI(X;Y)}$ distinguishable sequences of length n. Therefore,

$$\mathcal{L}(h_\Psi) = E_{(x,y)\sim P_{X,Y}} \ell\big(h_\Psi(y), x\big) = \sum_{x,y} P_{X,Y}(x, y) \ell\big(h_\Psi(y), x\big) = \sum_{x,y \in B} P_{X,Y}(x, y) \ell\big(h_\Psi(y), x\big),$$

where the last equation follows from (29). If $m \geq 2^{nI(X;Y)}$, and since $|B| = 2^{nI(X;Y)}$, and $\Pr[(x^n, y^n) \in B] = 2^{-nI(X;Y)}$, then clearly $\mathcal{L}(h_\Psi) = \mathcal{L}_\Psi(h_\Psi) \leq \Delta_m$. Otherwise, when $m < 2^{nI(X;Y)}$, we have

$$\mathcal{L}(h_\Psi) \leq (2^{nI(X;Y)} - m)2^{-nI(X;Y)} + m\Delta_m 2^{-nI(X;Y)} \leq \Delta,$$

assuming $\ell(x, \hat{x}) \leq 1$. We now have

$$m2^{-nI(X;Y)}(1 - \Delta_m) \geq 1 - \Delta, \tag{32}$$

Assuming $1 - \Delta_m \approx 1$,

$$m \geq 2^{nI(X;Y)}(1 - \Delta). \tag{33}$$

Equivalently

$$\Delta \geq 1 - \frac{m}{2^{nI(X;Y)}}(1 - \Delta_m). \tag{34}$$

$\square$

## C  RECEPTIVE FIELD NORMALIZATION

Pereg et al. (2021) propose a fast alternative algorithm, inspired by the classic iterative thresholding algorithms, that produces a relatively good approximation of a convolutional sparse code. Most solvers are slowed down by the use of one global threshold (bias) to detect each local feature shift along the signal, or a predetermined constant local threshold. When we apply a global threshold at each iteration, if the threshold is too high, weak expressions are annihilated, and strong expressions can "shadow" over low-energy regions in the signal, which in turn, can be interpreted as false-positive support locations. On the other hand, if the threshold is very small, as often is the case in ISTA, many iterations are required to compensate for false detections of early iterations. This issue is aggravated in the presence of noise, and in real-time applications due to model perturbations. The proposed remedy is to normalize each data point by a locally focused data energy measure, before applying a threshold. In other words, each receptive field of the data is scaled with respect to the local energy. This way even when the data is inherently unbalanced, we can still use a common bias for all receptive fields, without requiring many iterations to detect the features support.

We briefly reformulate RFN for 2D signals.

*Definition C.1 (2D Receptive Field Normalization Kernel)*: A kernel $h[k, l]$, $k, l \in \mathbb{Z}$, can be referred to as a receptive field normalization kernel if

1. The kernel is positive: $h[k, l] \geq 0 \quad \forall k, l$.
2. The kernel is symmetric: $h[k, l] = h[-k, -l] \quad \forall k, l$.
3. The kernel's global maximum is at its center: $h[0, 0] = 1 \geq h[k, l] \quad \forall k, l \neq 0$.
4. The kernel's energy is finite: $\sum_{k,l} h[k, l] < \infty$.

*Definition C.2 (2D Receptive Field Normalization)*: We define the local weighted energy centered around $S[k, l]$, a sample of a 2D observed signal $\mathbf{S} \in \mathbb{R}^{L_r \times J}$,

$$\sigma_S[k, l] \triangleq \left( \sum_{k', l' = -\frac{L_h - 1}{2}}^{\frac{L_h - 1}{2}} h[k', l']S^2[k - k', l - l'] \right)^{\frac{1}{2}}, \tag{35}$$

where $h[k, l]$ is a RFN window function of size $L_h \times L_h$, $L_h << L_r$ is an odd number of samples. For our application we used a truncated Gaussian-shaped window, but one can use any other window function depending on the application, such as: a rectangular window, Epanechnikov window, etc. The choice of the normalization window and its length affects the thresholding parameters. If $h[k, l]$ is a rectangular window, then $\sigma_S[k, l]$ is simply the Frobenius norm of a data patch centered around the $[k, l]$th location. Otherwise, if the chosen RFN window is attenuating, then the energy is focused in the center of the receptive field, and possible events at the margins are repressed. RFN is employed by dividing each data point by the local weighted energy, before projecting the signal on the dictionary and taking a threshold. We compute local weighted energy as defined in (35). Namely,

$$\sigma_S[k, l] = \sqrt{h[k, l] * S^2[k, l]}, \tag{36}$$

where $h[k, l]$ is a receptive field normalization window, and $*$ denotes the convolution operation. Then, we normalize the signal by dividing each data point by the corresponding receptive field energy. In order to avoid amplification of low energy regions, we use a clipped version of $\sigma_S[k]$. Namely,

$$\tilde{\sigma}_S[k, l] = \begin{cases} \sigma_S[k, l] & |\sigma_S[k, l]| \geq \tau \\ 1 & |\sigma_S[k, l]| < \tau \end{cases}, \tag{37}$$

where $\tau > 0$ is a predetermined threshold. Empirically, for our application $0.15 \leq \tau \leq 0.4$ works well. The RFN image $\tilde{\mathbf{S}}$ image is

$$\tilde{\mathbf{S}}[k, l] = \frac{\mathbf{S}[k, l]}{\tilde{\sigma}_{\mathrm{S}}[k, l]}. \tag{38}$$

Assuming an analysis patch $\{\mathbf{y}_i\}_{i=1}^{JL_{\mathrm{r}}} \in \mathbb{R}^{L_{\mathrm{t}} \times N}$ from RFN image $\tilde{\mathbf{S}} \in \mathbb{R}^{L_{\mathrm{r}} \times J}$ (as defined in Definition 5), given a normalized convolutional dictionary $\mathbf{D}$, an initial solution $\mathbf{x}_0 = \mathbf{0}$, the detected support (at the first iteration) is

$$\mathbf{q}_i = \mathcal{I}_{\beta_1}(\mathbf{D}^T \mathbf{y}_i), \tag{39}$$

where $\mathcal{I}$ is an element-wise thresholding indicator function

$$\mathcal{I}_{\beta_1}(x_k) = \begin{cases} 1 & |x_k| \geq \beta_1 \\ 0 & |x_k| < \beta_1. \end{cases} \tag{40}$$

---

**Algorithm 1:** 2D RFN-RNN (forward pass)

---

**input** : Input image $\mathbf{S} \in \mathbb{R}^{L_{\mathrm{r}} \times J}$, RFN-kernel $\mathbf{h} \in \mathbb{R}^{L_{\mathrm{h}} \times L_{\mathrm{h}}}$, $\mathbf{W}_{zy}$, $\mathbf{W}_{zz}$, $T$
**Init:** $\mathbf{z}_0 = \mathbf{0}$, $\mathbf{q}_0 = \mathbf{0}$, $\theta = 0$
compute:

$$\sigma_{\mathrm{S}}[k, n] \triangleq \sqrt{h[k, l] * S^2[k, l]}.$$

$$\tilde{\sigma}_{\mathrm{S}}[k, l] = \begin{cases} \sigma_y[k, l] & |\sigma_{\mathrm{S}}[k, l]| \geq \tau \\ 1 & |\sigma_{\mathrm{S}}[k, l]| < \tau \end{cases},$$

$$\tilde{S}[k, l] = S[k, l] / \tilde{\sigma}_{\mathrm{S}}[k, l]$$

**for** $k$ to $L_{\mathrm{r}}$ **do**
  **for** $l$ to $J$ **do**
    set: analysis patch $\{\mathbf{y}_i\}_{i=1}^{JL_{\mathrm{r}}} \in \mathbb{R}^{L_{\mathrm{t}} \times N}$ from image $\mathbf{S} \in \mathbb{R}^{L_{\mathrm{r}} \times J}$, and its corresponding
    analysis patch $\{\tilde{\mathbf{y}}_i\}_{i=1}^{JL_{\mathrm{r}}} \in \mathbb{R}^{L_{\mathrm{t}} \times N}$ from RFN image $\tilde{\mathbf{S}} \in \mathbb{R}^{L_{\mathrm{r}} \times J}$
    **for** $t = 1$ to $T$ **do**
      $\mathbf{q}_t = \sigma(\mathbf{W}_{zy}^T \tilde{\mathbf{y}}_t + \mathbf{W}_{zz}^T \mathbf{z}_{t-1} + \mathbf{b})$
      $\mathbf{r}_t = c_1 \mathbf{W}_{zy}^T \mathbf{y}_t$
      $\mathbf{z}_t = \mathbf{q}_t \odot \mathbf{r}_t$
    **end**
  **end**
**end**

---

The implied assumption is that $\mathbf{W}_{zy}$ represents $\frac{1}{c}\mathbf{D}$, therefore, it is also possible to add additional regularization loss terms enforcing $\mathbf{y}_t = \mathbf{D}\mathbf{z}_t$, $\tilde{\mathbf{y}}_t = \mathbf{D}\tilde{\mathbf{z}}_t$, where $\tilde{\mathbf{y}}_t$ is the RFN-input signal and $\tilde{\mathbf{z}}_t$ is the RFN latent supports vector.

## D EXPERIMENTS DETAILS

### D.1 IMAGE DEBLURRING

Table 1: PSNR (dB) and SSIM scores for natural image deblurring

| | Butterfly | Boats | C.Man | House | Parrot | Lena | Barbara | Starfish | Peppers | Leaves | Avg |
|---|---|---|---|---|---|---|---|---|---|---|---|
| Input PSNR | 22.84 | 26.40 | 23.30 | 28.52 | 27.05 | 26.33 | 23.84 | 24.80 | 25.97 | 22.26 | 25.26 |
| FISTA | 30.36 | 29.36 | 26.81 | 31.50 | 31.23 | 29.47 | 25.03 | 29.65 | 29.42 | 29.36 | 29.22 |
| NCSR | 30.84 | 31.49 | 28.34 | 33.63 | 33.39 | 31.26 | 27.91 | 32.27 | 30.16 | 31.57 | 31.09 |
| RED:SD-TNRD | 31.57 | 31.53 | 28.31 | 33.71 | 33.19 | 31.47 | 26.62 | 32.46 | 29.98 | 31.95 | 31.08 |
| RNN | 27.80 | 29.40 | 25.44 | 31.27 | 30.25 | 29.00 | 27.75 | 29.25 | 28.17 | 26.82 | 28.52 |
| | 0.93 | 0.91 | 0.88 | 0.93 | 0.95 | 0.92 | 0.82 | 0.92 | 0.90 | 0.94 | **0.91** |
| RNN-RFN | 26.16 | 28.46 | 25.16 | 30.65 | 29.81 | 28.49 | 24.47 | 28.73 | 28.01 | 25.61 | 27.56 |
| | 0.91 | 0.90 | 0.88 | 0.92 | 0.94 | 0.91 | 0.81 | 0.92 | 0.91 | 0.91 | **0.90** |

To test the systems' performance under more challenging setting we tested RNN and RNN-RFN with increasing additive noise variance $\sigma_n$ in the range $[0, 10\sqrt{2}]$. We trained the network with a

*single* example of the blurred image boat contaminated by Gaussian noise with variance $\sigma_n = \sqrt{2}$. As can be seen in Fig. 6 as the noise level is increased - RFN suppresses noise better. Figures 7-8 provide additional examples of the reconstruction of image starfish for $\sigma_n = 2\sqrt{2}$ and $\sigma_n = 7\sqrt{2}$.

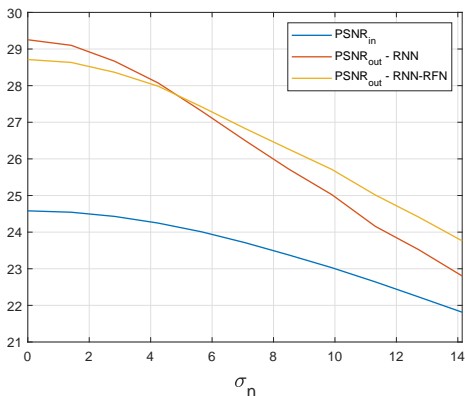

Figure 6: PSNR[dB] as a function of number of noise variance $\sigma_n$.

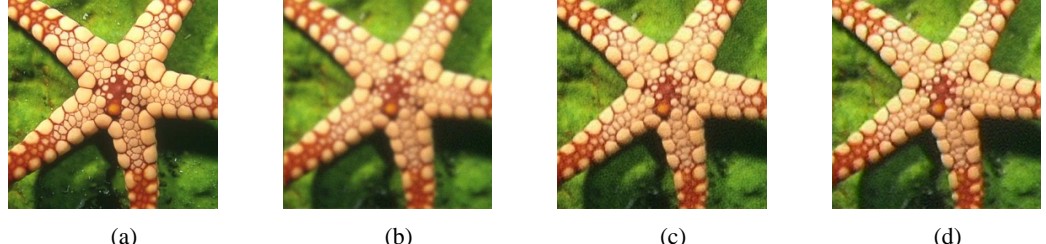

| (a) | (b) | (c) | (d) |

Figure 7: Visual comparison of deblurring of the image starfish for different noise levels: (a) Ground truth; (b) input, PSNR=24.25dB, $\sigma_n = 4.24$; (c) RNN result, PSNR=28.07dB ; (d) RNN-RFN result, PSNR=27.91dB.

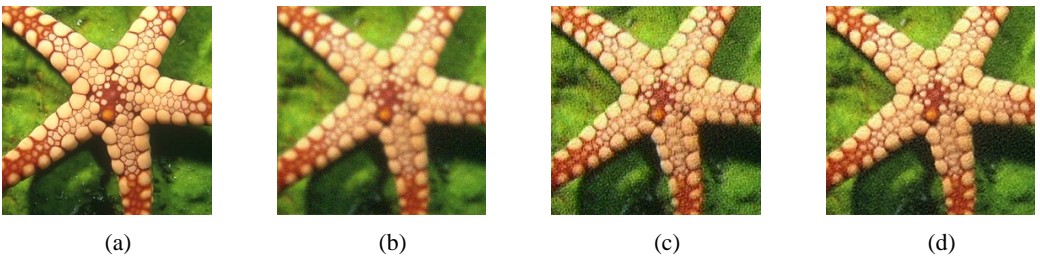

| (a) | (b) | (c) | (d) |

Figure 8: Visual comparison of deblurring of the image starfish for different noise levels: (a) Ground truth; (b) input, PSNR=23.02dB, $\sigma_n = 9.90$; (c) RNN result, PSNR=25.02dB ; (d) RNN-RFN result, PSNR=25.07dB.

## D.2 OCT SPECKLE SUPREESION

The learning system is the RNN framework described in section 5. We denote $\mathbf{S} \in \mathbb{R}^{L_r \times J}$ as a cropped single intensity image of a coherent tomogram aligned with a speckle suppressed tomogram $\mathbf{R} \in \mathbb{R}^{L_r \times J}$. We then define an analysis patch as a 2D patch of size $L_t \times N$ enclosing $L_t$ time (depth) samples of N consecutive traces of the observed image $\mathbf{S}$. An analysis patch $\mathbf{A}^{(i,j)}$ associated with a pixel $\mathbf{R}_{i,j}$ in the speckle suppressed intensity tomogram. To produce a point in the final tomogram $\mathbf{R}_{i,j}$, we set the input to the RNN as $\mathbf{y} = \mathbf{A}^{(i,j)}$. Each time step input is a group of $N$ neighboring pixels of the same corresponding time (depth). In other words, $\mathbf{y}_t = [\mathbf{S}_{i-L_t+1+t,j-n_L}, ..., \mathbf{S}_{i-L_t+1+t,j+n_R}]^T$ is row $t$ of the corresponding analysis patch. We set the size of the output vector $\mathbf{x}_t$ to one expected pixel: $P = 1$. Such that $\mathbf{x}$ is the expected segment

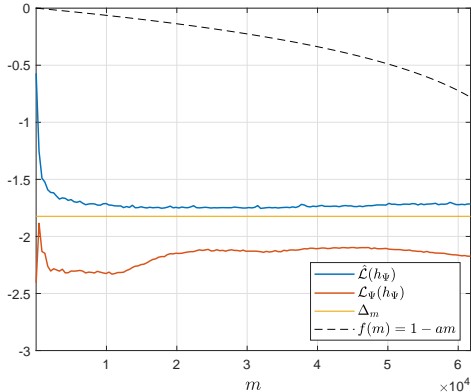

Figure 9: Recovery error $\hat{\mathcal{L}}(h_\Psi)$ and training error $\mathcal{L}_\Psi(h_\Psi) \leq \Delta_m$ ($\log_{10}$ scale) as a function of the sample size $m$. Empirically, $\mathcal{L}_\Psi(h_\Psi) < 1 - am$, where $a = 2^{-nI(X;Y)}(1 - \Delta_m)$, for $I(X;Y) \geq 0.33$.

of the corresponding enhanced tomogram, $\mathbf{x} = \left[ \mathbf{R}_{i-(L_\mathrm{t}-1),j}, ..., \mathbf{R}_{i,j} \right]^T$. Lastly, we ignore the first $L_\mathrm{t} - 1$ values of the output $\mathbf{x}$, and set the predicted pixel $\hat{\mathbf{R}}_{i,j}$ as the last one, $\mathbf{x}_{L_\mathrm{t}}$

The analysis patch moves through the image and produces all predicted speckle suppressed intensity tomogram points in the same manner. Each output pixel is recovered by taking into consideration its relations to neighboring pixels: $L_{\mathrm{t}-1}$ preceding pixels in the axial direction at the same lateral position, and $(N-1)L_\mathrm{t}$ pixels from $N-1$ laterally neighboring surrounding pixels (in space), from adjacent A-lines. For all experiments, we set the number of neurons as $n_n = 1000$. Increasing the number of neurons did not improve the results significantly, but increases training time.

In a recurrent neural network the nodes of the graph are connected by both feedforward connections and *feedback* connections (Hochreiter & Schmidhuber, 1997; Géron, 2017): At a current state, the current output depends on current inputs, and on outputs at previous states. In a sense, we can say that the network is able to make decisions that are based also on the memory of its past decisions, that it "remembers" its past outputs and takes them into consideration when computing the current output. We believe RNN fits this task because it is able to "remember" both in space and time dimensions. Most OCT images have layered structure, and exhibit string relations along the axial and lateral axes. RNNs can efficiently capture those relations and exploit them.

Figures 4(a)-(b) show the speckle-corrupted and speckle-suppressed tomograms of chicken tissue of size $341 \times 691$ used for training. The analysis patch size was of size $[L_t, N] = [9, 9]$. The axial and lateral spacing for both training and testing data are $\Delta_\mathrm{x} = 3.06\mu\mathrm{m}, \Delta_\mathrm{z} = 6\mu\mathrm{m}$.

In Figs. 5(a)-(b) a tomogram of size $300 \times 643$ was used for training. Here, the lateral and axial spatial sampling spacing are $\Delta_\mathrm{x} = 2.5\mu\mathrm{m}, \Delta_\mathrm{z} = 4.78\mu\mathrm{m}$, respectively. Whereas, the cucumber axial and lateral spacing are $\Delta_\mathrm{x} = 8\mu\mathrm{m}, \Delta_\mathrm{z} = 4.78\mu\mathrm{m}$, respectively. Note that, the cucumber image is sampled at significantly lower rate along the lateral direction. Due to the system mismatch between the training data and the testing data, we set the analysis patch size to [8,7], to match the testing data parameters, rather than the optimal analysis patch size that would fit the training data.

