# OpenReview forum: "Less is More: Rethinking Few-Shot Learning and Recurrent Neural Nets"
_ICLR.cc/2023/Conference — Submitted to ICLR 2023_

### Official Review · Reviewer_WHvp · 2022-10-19

**Confidence:** 3
**Correctness:** 2
**Technical Novelty And Significance:** 3
**Empirical Novelty And Significance:** 3
**Recommendation:** 3

**Clarity, Quality, Novelty And Reproducibility:**

I think the paper is original, but a clearer connection among sections and better explanation would be helpful.

**Strength And Weaknesses:**

This paper is broad that touches three aspects of the learning problem, and the insight from info theory provides a different perspective to ML community. I believe the math proof is correct and there are illustrative experiments as well. Below are detailed questions.

- The introduction is not clear. I would suggest moving Figure 1 to the front. Now when I read it, seems the beginning part starts from introducing the last two blocks of the figure, not mentioning the left part. I’m confused about the setup yet, is it like, there is a system that encodes x as y, and we want to find the decoder/inverse map from y to x? How is it relevant to RNN?
- I think the logic is not quite clear to readers. The connection among three parts seems to be weak. For example, the first part is relevant to sample complexity, but the second part is analysis of RNN forward algorithm. How is it relevant to typical set stuff?
Although there are a few theorems in the first part, I think most of them are textbook knowledge. The entropy is directly relevant to how hard the learning is and the typical set represents the most likely appearing instances following the distribution. I don’t find much novelty here.
- The justification of the Algo 1, especially the novelty, is very short and I cannot understand it.
- The paper has the title “few-shot”, but I cannot see how this paper contributes to it, i.e. how to learn from “few” data. The typical set stuff seems to be relevant, but perhaps I did not catch its relation to the other sections.
- Section 3,4 constructs the equivalence of iterative lasso solver and RNN, but I cannot see why this connection is important and correct. For example, the whole sequence of RNN is important, but are the iterations in the middle of lasso solver important, what do they mean? I feel in lasso, the iterations would be more and more sparse, is it an expected and important property of RNN sequence? Slight differences between lasso and RNN exist, like the bias $b$, and the thresholding parameter $\lambda$, etc.
- Critical: Comparing (14) and (15), is $W_{zz}$ forced to be a PD matrix as assumed in lasso solver, is it enforced that $I -c W_{zy}W_{zy}^T = W_{zz}$? If $W_{zy}$ and $W_{zz}$ represent $D^T/c$ and $I - D^TD/c$ then they have to follow that equality. One needs to justify and properly use the similarity between lasso solver and RNN, otherwise it just looks like a coincidence.
- Page 5, $cI-D^TD \succ 0$.

**Summary Of The Paper:**

This paper analyzes the learning problem in three aspects. 1. From the perspective of information theory, when the loss is bounded and the entropy of the data distribution is small, one can learn only in the typical set; 2. Modeling RNN propagation as the iterative method that solves sparse sensing and RNN training is equivalent to dictionary learning; 3. Proposed the RNN algorithm with weighted image. Finally there are experiments verifying the points.

**Summary Of The Review:**

Given my confusions and questions above, I would like to see more authors' response and clarification, and would definitely be willing to increase the score if it's convincing.

---

> ### Author Response · Authors · 2022-11-17
> **Reply to reviewer WHvp (part A)**
>
> Re Summary Of The Paper:
> From the perspective of information theory (Section 2), we highlight three points:
>
> (1) Learning from the typical set is equivalent to learning from the data distribution in terms of generalization error (Theorem 3).
>
> (2) When learning is in a noiseless environment the sample size and the generalization error, of a specific trained system with training error \Delta_m, obey the relationship described in equation (3). In words, as long as the sample size is larger or equal to the size of the typical set, we are guaranteed to have a generalization error that is as small as our training error. Otherwise, generalization error lower bound is determined by the ratio between the sample size and the typical set (see also equations (27)-(28)).
>
> (3) When learning in a noisy environment the sample size and the generalization error, of a specific trained system with training error \Delta_m, obey the relationship described in equation (4). As explained in the paper, in this case the generalization error lower bound is determined by the sample size and the input-output mutual information (see also equation (33)-(34)).
> This is true whether the entropy is large or small, but obviously would permit more efficient learning in cases where the entropy is small.
>
> We have corrected the explanations in section 2 and in the proofs (Appendix B) to make these points clearer to the readers.
>
> Re Strength And Weaknesses:
>
> 1. The last two blocks are the assumed learning system (not necessarily built as an encoder-decoder, but true for the RNN framework, and many other possible architectures). The first two blocks are usually more relevant for inverse problems, as stated in the paper: “Let us assume a training set $\Psi=$ {\mathbf{y}_{i},\mathbf{x}_{i} _{i=1}^m\}, where $\mathbf{x}_i \in \mathbb{R}^{n\times 1}$ are sampled from $A^n_\epsilon(P_X)$ and paired with $\mathbf{y}_i \in \mathbb{R}^{d\times 1}$ by some function as ground truth. The learning system is trained to output a prediction rule $\mathcal{F}: \mathcal{Y}^d \rightarrow \mathcal{X}^n$. Assume an algorithm that trains the predictor by minimizing the training error (empirical error or empirical risk). Assuming a discrete source $P_X(\cdot)$ that emits i.i.d sequences $x^n$ of symbols (for example: patches in an image, segments of an audio signal, etc.), the estimator has access only to the observed signal $y^d$. In an inverse problem $y^d$ would be a degraded signal originating in $x^n$, where the relationship between $x^n$ and $y^d$ could be linear or non-linear, with or without additive noise, such that generally, $y^d=g(x^n)+e(x^n)$, where $g(\cdot)$ and $e(\cdot)$ are functions of $x^n$. Given $y^d$, we produce an estimate $\hat{x}^n$. However, the following proofs are not restricted to this framework. Our problem setting is illustrated in Fig. 1.”
> This part is not yet related to RNNs. The RNN frameworks serve as an empirical example to demonstrate learning from a smaller set. Or alternatively, the theoretical part suggests an explanation for the results of the experimental part.
> We also show that RNNs are sparse solvers, but this is also unrelated to the AEP theory. In the RNN framework described in section 3.1, the RNN is the encoder, and the FC network is the decoder.
>
> As stated in the image deblurring section 6.1: “Hence, it is safe to assume that the patches in these images reliably represent the typical set associated with natural images to a certain degree.” That is, one could hypothesize that learning from one image is possible due to the existence of a relatively small set that reliably represents the entire data distribution.
>
> We had to keep figure 1 in its original place to comply with the 9 pages limit.

---

> > ### Comment · Reviewer_WHvp · 2022-11-22
> > **Thanks for the comment**
> >
> > The explanation is same as what I interpreted the paper, thanks.

---

> ### Author Response · Authors · 2022-11-17
> **Reply to reviewer WHvp (part B)**
>
> 2. The analysis of RNN as a sparse solver is unrelated to the typical set and sample complexity part. However, both parts are related to the RNN framework used for few-shot learning. We do not imply that the interpretation of an RNN as a sparse solver in connected to the AEP theory for ML.
>
> As far as we know, theorems 3-5 are completely novel. As of today, to the best of our knowledge there is no textbook discussing the relationship between AEP theory to ML and its practical implications. The insights in the papers are completely novel.
> The theoretical part is closely related to the experimental part. To more formally explore this connection, we added Figure 9 and the corresponding text: ”As the focus of this study is to bridge between theory and application for few-shot learning, and to provide theoretical justifications, rather than to achieve state-of-the-art results, we verified the theoretical results described in Theorem 4 in practice for the image deblurring case. To this end, we trained the RNN network with different sample sizes while trying to keep a constant training error,  $\mathcal{L} \leq \Delta_{m}$. The training data consists of patches belonging to a single image (starfish). We consider the recovery error, denoted as $\hat{\mathcal{L}}(h_{\Psi})$,
> over the other non-training 9 images as an empirical approximation for the generalization error. Figure 9 shows the evolution of the recovery error with increasing sample size chosen uniformly in the range [1, 61752].Evidently, $\hat{\mathcal{L}}(h_{\Psi}) < \mathcal{L} $ while $\mathcal{L}\geq\Delta_m$ ,which implies $m<2^{mI(X;Y)}$. In this region $\hat{\mathcal{L}}(h_{\Psi})  \leq 1 - m 2^{-nI(X;Y)}(1-\Delta_m) \leq \Delta$ (see proof). Hence, empirically we can deduce ${I(X;Y)} \geq 0.33$.”
> (I replaced the notation for the training error here due to in the Latex  compiler)
>
> 3. As stated in the paper, receptive-field-normalization (RFN) was first proposed by (Pereg et al., 2021) for a speed-up version of a sparse coder for seismic inversion. We simply extend its use for RNNs. RFN (Pereg et al., 2021) refers to local signal normalization, in an attempt to find the detected features’ support in fewer iterations, by avoiding the liability of unbalanced energy. Since the RNN is essentially a sparse solver, we propose to apply 2D-RFN, prior to applying
> the RNN. We then multiply the latent space support normalized weights by the non-normalized projection of the learned dictionary on the original input signal to regain the local energy.
>
> As stated in the paper, from the theoretical point of view of the paper (the AEP-ML theory), RFN signal’s distribution is confined to a smaller set of values (because all local maxima absolute values are mapped to values around 1). Thus, RFN decreases the signal’s entropy and therefore the size of the typical set associated with the input distribution. Practically, as mentioned in the experimental results section, RNN-RFN has an advantage in terms of visually suppressing noise (but with similar qualitative scores).
>
> We also added Figure 6-8 and the corresponding text: “Figure 6 show the evolution of the PSNR scores for RNN and RNN-RFN with increasing additive noise variance $\sigma_n \in [0,10\sqrt{2}]$. As the noise level is increased RFN suppresses noise better. Additional examples and training details are in Appendix D.1.”
> For the sake of brevity, we had to keep the relevant explanations short. More details can be found in Appendix C and in [1].
>
> 4.Reply: Section 2 of the papers explains why in theory there exists a smaller set that can reliably represent the data distribution and can be used for learning.
> In sections 5-6 we show in practice and examples, where a system is trained with patches of one image and is surprisingly able to generalize well.
> “Hence, it is safe to assume that the patches in these images reliably represent the typical set associated with natural images to a certain degree.”

---

> > ### Comment · Reviewer_WHvp · 2022-11-22
> > **Comment**
> >
> > Regarding the typical set stuff, I think the definition of typical set is a subset of all inputs that happens with high probability. As the expected loss is probability and loss value product, if loss value is bounded, then a subset with small probability has little affect to the total expected loss, which is as expected.
> >
> > I think the AEP part means a distribution with less entropy can be represented with fewer data in the typical set. Could you explain more how RFN makes the entropy small?

---

> > > ### Author Response · Authors · 2022-11-23
> > > **RFN reduced entropy**
> > >
> > > Receptive field normalization locally normalizes each point in the center of a receptive field (an image patch), by the local energy around that point in the signal. The result is the receptive-field-normalized signal, where local minima and maxima are mapped to values +-1 (approximately), and the rest of the values are in the range [-1,+1].
> > > Figure 1 in (Pereg et al., 2021) provides a clearer visualization of this phenomenon.
> > > (see here
> > > https://israelcohen.com/wp-content/uploads/2021/08/Convolutional_Sparse_Coding_Fast_Approximation_with_Application_to_Seismic_Reflectivity_Estimation-2.pdf)
> > > As can be seen in this figure, local maxima are mapped to a value around one (or a different maximum value close to one, depending on the chosen parameters). (For images in the range [0,255] RFN is applied to the input image to the neural net that is shifted and rescaled to the range [-0.5, 0.5]).
> > > Once all local maxima/minima can no longer have different values, and all the other different values in the signal are also mapped to a shared range, the new RFN images' pdf (probability distribution function) is now confined to a smaller set of values.
> > > Hence, the uncertainty of the RFN signal is lower.
> > > And so, without estimating the image patches' entropy (which is complicated, and requires an estimation of the pdf, thus requiring many more patches), we can still guarantee the RFN signal's entropy is smaller than the original signal's entropy. Therefore, the typical set that is associated with that signal is smaller. (This is pretty much very intuitive and probably clearer with the illustration in Figure 1 of (Pereg et al., 2021)). For a uniform receptive field normalization kernel $||\mathbf{x}_i||_2 \geq ||\mathbf{x}||_\infty= \max_i x_i $ where $x_i$ is a point (pixel) value and $||\mathbf{x}_i||_2$ is the local receptive field energy (patch $\ell_2$ norm) around that point.
> > >
> > > As we briefly stated in the paper:
> > > "The RFN signal's distribution is confined to a smaller set of values. Thus, RFN decreases the signal's entropy and the size of the typical set associated with the input distribution."
> > >
> > > Please let me know if this explanation is clear, or if further clarification is required.
> > >
> > > Thanks a lot !!

---

> ### Author Response · Authors · 2022-11-17
> **Reply to reviewer WHvp (part C)**
>
> 5. As stated in the paper: “Clearly, (15) is equivalent to (5). In other words, a RNN can be viewed as unfolding of one iteration of a learned sparse coder, based on an assumption that the solution at time $t$ is close to the solution in time $t-1$.
> In subsection 3.1, the RNN state encodes the vector to a latent space.
> Then the FC net decodes the latent variable back to a space of the
> required dimensions.
> Given this interpretation, it may be claimed that RNN's use should not be restricted to data with obvious time or depth relations. The RNN merely serves as an encoder providing a rough estimation of the sparse code of the input data. The RNN's memory serves the optimization process by starting the computation from a closer solution. Thus placing the initial solution in a ``close neighborhood" or close proximity, and helping the optimization gravitate more easily towards the latent space sparse approximation. Clearly, convergence is not guaranteed.”
>
> The whole sequence of RNN is not compared to the middle iterations of the LASSO solver. Each time-step approximates a sparse code in one iteration, starting with an initial solution that is perhaps closer. We do not imply that each time-step would make the solution more and more sparse. The purpose of this section is to provide a brief intuition for RNN operation, without further analysis.
>
> 6. $W_{zz}$ is not forced to be positive-definite. LISTA (Gregor & Lecun, 2010) uses similar logic, as well as many other algorithm-unrolling methods (V. Monga, Y. Li and Y. C. Eldar, "Algorithm Unrolling: Interpretable, Efficient Deep Learning for Signal and Image Processing," in IEEE Signal Processing Magazine, vol. 38, no. 2, pp. 18-44, March 2021, doi: 10.1109/MSP.2020.3016905), without further justification. The similar structure may have been a coincidence, to begin with, nevertheless, it still has meaning and insight.
>
> 7. Reply: corrected. Thank you !!
>
> Re Clarity, Quality, Novelty And Reproducibility, Summary:
> We made all the necessary corrections advised by the reviewer to improve clarity and quality and replied to the reviewer’s comments regarding novelty and reproducibility. We hope that the reviewer could reconsider the paper for publication.  We would like to thank you and the anonymous reviewers for the valuable comments and helpful suggestions. A revised version of the referenced manuscript has been submitted online. The responses to the reviewers’ comments are summarized above.

---

> > ### Comment · Reviewer_WHvp · 2022-11-17
> > **Critical**
> >
> > First sorry for my bad, I meant that $I - W_{zz}$ needs to be PSD since $W_{zz}$ represents $I - D^TD/c$.
> >
> > As I mentioned in the previous part "critical". I believe the point that "$W_{zy}$ and $W_{zz}$ represent $D^T/c$ and $I - D^TD/c$" is critical for the comparison. If the comparison is true, then you must have $I -c W_{zy}W_{zy}^T = W_{zz}$. Otherwise the sequence cannot be seen as the iterations of the sparse recovery problem and the claimed similarity does not exist.
> >
> > The RNN architecture can be intrinsically seen as unrolling *something*, but the question "what is that *something*" needs to be carefully reasoned about.
> >
> > This is about the *correctness* of the section so it is very critical. If the other paper justifies the same point (that I did not find), please quote that. Otherwise I would say the other paper is incorrect as well -- I do not believe a paper has to be correct only if it's published anywhere -- there has to be a justification. The authors can also consider removing the section. I believe *correctness* is the baseline of a paper (there is not much time left for editing the paper so I reply to this comment first).

---

> > > ### Author Response · Authors · 2022-11-17
> > > **Correctness of Section 4**
> > >
> > > Of course, it is agreed that correctness is very important. And we should definitely remove the section if it is incorrect. But I also don't think we should remove it if it is very likely is correct.
> > >
> > > Note that we follow the exact steps that lead from ISTA to LISTA. I'm sorry but I don't see how did they force $I-W_{zz}$ to be positive-definite. I don't just assume its correctness due to the fact it is published. Gregor & Lecun's work on LISTA is a seminal work. Also quoted by
> > > Papyan, V., Romano, Y. and Elad, M., 2017. Convolutional neural networks analyzed via convolutional sparse coding. The Journal of Machine Learning Research, 18(1), pp.2887-2938.
> > > where they say something very similar: "In (Gregor and
> > > LeCun, 2010) it was shown that the IST algorithm can be formulated as a simple recurrent
> > > neural network. " (page 25)
> > >
> > > Perhaps the misunderstanding here, in my opinion, is that it is *not expected nor required* that $W_{zz}$ would converge to $\frac{1}{c}\mathbf{D}^T$ and for $W_{zy}$ to converge to $I-\frac{1}{c}\mathbf{D}^T\mathbf{D}$, as in a dictionary learning task. It definitely wasn't proved that the trained network weights obey that.
> > > Practically, it is possible to implement that (by adding loss terms). And I have also experimented with that, though I didn't notice any substantial effect on the results.
> > >
> > > In terms of correctness, since we are not claiming that equation (15) would end up equivalent to equation (14), after training, I don't think it is fair to say that our proposed intuition is wrong. In the same way that LISTA doesn't claim that equation (11) is equivalent to equation (10). LISTA's paper is named Learning Fast Approximations of Sparse Coding, and they have demonstrated that the algorithm reliably approximates sparse coding.
> > >
> > > I think that rather than removing the section altogether, perhaps the reviewer could consider rephrasing the text in a manner that doesn't mislead the reader. We did not intend to imply that the trained network would be the exact implementation of dictionary learning for ISTA, and we have not written that in the text.
> > >
> > > As this is time sensitive, please let me know your thoughts as soon as possible.
> > >
> > > Thanks a lot !!

---

> > > > ### Comment · Reviewer_WHvp · 2022-11-17
> > > > **Follow-up**
> > > >
> > > > I agree that IST algorithm can be seen as a *special* RNN, but not the other direction. I don't think one can interpret any RNN as ISTA since the relationship between $W_{zz}, W_{zy}$ does not follow. But if you can enforce the connection and do not see a difference in practice, I would like to enforce it to make it in accordance.
> > > >
> > > > If "we are not claiming that equation (15) would end up equivalent to equation (14)", then I suggest the authors considering revising sentence between (14) and (15)
> > > >
> > > > "which in its learned version can be *replaced* by"
> > > >
> > > > and the sentence below (15)
> > > >
> > > > "a RNN can be viewed as unfolding of one iteration of a learned *sparse coder*"
> > > >
> > > > and add a thorough explanation regarding this transition. If $W_{zz}, W_{zy}$ does not follow the aformentioned relation, then the RNN iterations cannot be seen as iterations of learning a *sparse* coder.

---

> > > > > ### Comment · Reviewer_WHvp · 2022-11-17
> > > > > **I'm not sure if asking in the following way is clearer...**
> > > > >
> > > > > The point is, you have two types of iterations:
> > > > >
> > > > > 1. general RNN architecture
> > > > > 2. learning sparse coder
> > > > >
> > > > > which look similar but have essential differences as well, which might give you totally different states/outputs. Then, why are they -- or to what extent are they -- still comparable, and what is the insight?
> > > > >
> > > > > I would say point 2 belongs to point 1 but not the other way around, which means you can explain 2 by using 1 but not explain 1 by using 2.

---

> > > > > > ### Author Response · Authors · 2022-11-17
> > > > > > **Follow up**
> > > > > >
> > > > > > I don't follow.
> > > > > > The RNN is not equivalent to ISTA, but it is 100% equivalent to one iteration of LISTA, with a non-zero initialization. And LISTA is defined as a *learned* sparse coder
> > > > > >
> > > > > > I'll try to rephrase it to the best of my understanding.

---

> > > > > > > ### Comment · Reviewer_WHvp · 2022-11-17
> > > > > > > **Comment**
> > > > > > >
> > > > > > > I'm not familiar with LISTA, from that paper I saw
> > > > > > >
> > > > > > > "the basic idea is to design a non-linear, parameterized, feed-forward architecture with a fixed depth that can be trained to approximate the optimal sparse code"
> > > > > > >
> > > > > > > is their target is to learn a sparse coder, and your target is to learn something (not have to be a sparse coder) by RNN? If so, there is still a difference -- If the targets are different, the learned parameters are different.
> > > > > > >
> > > > > > > Anyways I suggest making the paper self-constrained, the transition between (14) and (15), i.e., my point 1 and 2, needs to be justified. What are the same, what are different, and how can they be comparable.

---

> > > > > > > > ### Comment · Reviewer_WHvp · 2022-11-17
> > > > > > > > **For reviewer's easier reference**
> > > > > > > >
> > > > > > > > It would be helpful to distill and type the relevant justification/proof from another paper, and write it in the authors' comment as well as in the paper's appendix -- rather than a brief citation -- so we can see how the logic goes through.

---

> > > > > > > > ### Author Response · Authors · 2022-11-17
> > > > > > > > **Section 4 corrections**
> > > > > > > >
> > > > > > > > I uploaded a revised paper with some minor corrections.
> > > > > > > > (I adopted a similar some of the very brief summary of LISTA as in https://arxiv.org/pdf/2001.08456.pdf (Ada-LISTA, page 2, around equation (6). )
> > > > > > > >
> > > > > > > > Though in the original paper, LISTA is indeed trained to output sparse codes, they definitely don't end up with $\mathbf{W} = \frac{1}{c}\mathbf{D}^T$ nor  $\mathbf{S}=\big(\mathbf{I}-\frac{1{c}\mathbf{D}^T\mathbf{D}\big)$.
> > > > > > > > There is no way to prove the weights converge to the actual dictionary-related expression. In practice, they don't. Their case is mostly intuitive. And works in practice, but it is not justified by mathematical correctness (no positive-definiteness and so on...)
> > > > > > > > When LISTA is used as part of a system, where the output is not the latent space, then there is no direct training of that network to output sparse codes. Unfortunately as can be read in LISTA's paper, there is no "deeper" justification or proof for unfolding ISTA with renewed parametrization. They simply show a similar structure, reparametrize, and show results.
> > > > > > > >
> > > > > > > > As stated in Papyan's paper (page 3):
> > > > > > > > the forward pass of the CNN is in fact identical to our proposed pursuit – the layered
> > > > > > > > thresholding algorithm. This connection is of significant importance since it gives a clear
> > > > > > > > mathematical meaning, objective, and model to the CNN architecture, which in turn can
> > > > > > > > be accompanied by guarantees for the success of the forward pass, studied via the layered
> > > > > > > > thresholding algorithm." Similarly, there is no guarantee that the CNN's weights would converge to a certain specific dictionary.
> > > > > > > >
> > > > > > > > In our case Section 4 is general. One could be training the RNN to output sparse codes, or as a feature extractor, or as an encoder, or as one wishes. But the structural similarity exists. Similarity, and not equality.
> > > > > > > > When the RNN is used as an encoder, it is perhaps necessary to justify that its latent space code is sparse.
> > > > > > > > As stated in the paper in practice it is sparse. In my opinion, it might be related to our comment on over-parametrization, as the number of hidden units is higher, that would mean the number of features is increased, and naturally, there are "too many features" relative to the input dimension. (I redacted this intuition from the paper for the sake of brevity).
> > > > > > > >
> > > > > > > > I think the current phrasing: " a RNN can be viewed as an unfolding of one iteration
> > > > > > > > of a learned sparse coder, based on an assumption that the solution at time t is close to the solution
> > > > > > > > in time t − 1." is not misleading because it doesn't suggest that it is equivalent or identical. Just that this is a  possible intuition. A point of view. An interpretation. But not a proof, and no theorems.
> > > > > > > > I think the paper is self-contained. I don't think further analysis is necessary in the scope of this paper.
> > > > > > > > Further investigation is possible, but since this section is not the main focus of the paper, I refrained from further elaborating on this issue.
> > > > > > > >
> > > > > > > > Note that in Monga, V., Li, Y. and Eldar, Y.C., 2021. Algorithm unrolling: Interpretable, efficient deep learning for signal and image processing. IEEE Signal Processing Magazine, 38(2), pp.18-44."
> > > > > > > > they even mention that for LISTA: "In recent studies [22], [23],
> > > > > > > > [24], different parameters are employed in different layers" (page 4).
> > > > > > > > "As a particular example, in LISTA (see the box Learned ISTA),
> > > > > > > > the matrices Wt, We may be learned in each iteration, so
> > > > > > > > that they are no longer held fixed throughout the network.
> > > > > > > > Furthermore, their values may vary across different layers
> > > > > > > > rather than being shared. By allowing a slight departure
> > > > > > > > from the original iterative algorithms [13], [27] and extending
> > > > > > > > the representation capacity, the performance of the unrolled
> > > > > > > > networks may be boosted significantly."
> > > > > > > >
> > > > > > > > Finally, I would like to emphasize, that at no point in the paper, have we defined the RNN as identical to ISTA. It solely has a similar structure to LISTA. And as you mentioned, if trained with a direct sparse target, then it could be viewed as the exact equivalent of a single iteration of LISTA, with a different initialization. Otherwise, there are no guarantees, and I have explicitly phrased this part accordingly.

---

> > > > > > > > > ### Comment · Reviewer_WHvp · 2022-11-17
> > > > > > > > > **Thanks for your detailed comment**
> > > > > > > > >
> > > > > > > > > I think we have agreed on a few things: for the two points
> > > > > > > > >
> > > > > > > > > 1. general RNN architecture
> > > > > > > > > 2. learning sparse coder
> > > > > > > > >
> > > > > > > > > The iterations follow similar "structure", but the parameters ($D$ and $W$'s) are different.
> > > > > > > > >
> > > > > > > > > With all the comment being public here, I leave the question to other reviewers and AC: is it still a reasonable/valuable similarity between the two dynamics? I think the essence should be the similarity between the sequences generated by the two dynamics, but the sequences of 1 might not equal to a sequence of 2 with any $D$. So I do not think I'd change my position for now.
> > > > > > > > >
> > > > > > > > > I'm not sure if the following example illustrates my confusion:
> > > > > > > > >
> > > > > > > > > If I have a mapping $f(x) = Ax$, and $A$ is PSD, then I have the property "$\langle f(x), x \rangle \ge 0$". However, if I train a matrix $\hat A$ that $\hat f(x) = \hat A x$, I cannot claim that "$\langle \hat f(x), x \rangle \ge 0$", unless $\hat A$ is PSD. So that the "equality" of the parameters matters for maintaining/preserving the "positive inner product" property, or I can call it an insight or some meaning, though $f(x)$ and $\hat f(x)$ have similar "structures" i.e. both are linear mappings.
> > > > > > > > >
> > > > > > > > > I'm not totally sure whether Point 1 and 2 can generate similar sequences even they empirically have similarities in certain cases:
> > > > > > > > >
> > > > > > > > > - Especially, when a network is trained to output sparse codes, though it can end up with different parameters, the training target of Point 1 is same as Point 2 -- i.e., I want Point 1 to behave like Point 2, *in purpose* -- so that I think it reasonable that the sequences behave similarly.
> > > > > > > > >
> > > > > > > > > - On the other hand, it's hard to get an idea or insight how far/wide this phenomenon generalizes to all scenarios, when the RNN is trained to do all kinds of things, so a guideline in theory is important, maybe only for me.
> > > > > > > > >
> > > > > > > > > Afterall, the question is specifically about a part of the paper and the other sections can be insightful as well.

---

> > > > > > > > > > ### Author Response · Authors · 2022-11-18
> > > > > > > > > > **a few more comments**
> > > > > > > > > >
> > > > > > > > > > The sequence of 1 is definitely not equal to a sequence of 2 with any D, because for the RNN the solution is placed close to the previous time-step solution. A forward pass of LISTA with one iteration, and W=Wzy and S=Wzz, with an initial solution
> > > > > > > > > > $\mathbf{z}_{t-1}$,
> > > > > > > > > > would yield the exact same result by definition. But LISTA and ISTA are usually initialized with zero initialization $\mathbf{z}_0$ = $\mathbf{0}$,
> > > > > > > > > >
> > > > > > > > > > and therefore the one iteration with LISTA would simply be
> > > > > > > > > > $\mathbf{z}_{t}= \mathcal{S}_\beta \Big( \mathbf{W} \mathbf{y} \Big)$.
> > > > > > > > > > That is, simple thresholding of the projection of some learned dictionary on the signal (correlation detector).
> > > > > > > > > > Then, the second iteration of LISTA for sparse coding of the time-step $\mathbf{z}_t$ would be:
> > > > > > > > > >
> > > > > > > > > > $\mathbf{z}_{t} = \mathcal{S}_{\beta}  \Big( \mathbf{W} \mathbf{y} +  \mathbf{S} \mathbf{z}_{t-1} \Big)$
> > > > > > > > > >
> > > > > > > > > > (I have a bug here with the Latex compiler. I'm sorry).
> > > > > > > > > > Whereas for an RNN, as stated in the paper, there is only one thresholding step:
> > > > > > > > > > $\mathbf{z}_{t} = \mathcal{S}_{\beta}  \Big( \mathbf{W}^T_{yz} \mathbf{y} +  \mathbf{W}^T_{zz} \mathbf{z}_{t-1} \Big)$
> > > > > > > > > >
> > > > > > > > > > And so, these are definitely not equivalent, even with the same weights, and we are definitely not implying that they are the same.
> > > > > > > > > >
> > > > > > > > > > I understand your concerns regarding the equality of the parameters and the mathematical accuracy. Once the parameters are freed to be learned without imposing the D-link between W and S there is no guarantee that we follow the MM strategy, and no guarantee to converge correctly, as stated in the paper.  But the whole idea of letting the parameters be learned is that they are not controlled by the same constraints. The "best" parameters that fit the task are supposedly chosen during training.
> > > > > > > > > >
> > > > > > > > > > I have replaced ‘replace’ with ‘re-parameterize’ and added similar language just after equation 11.
> > > > > > > > > >
> > > > > > > > > > There is room for further theoretical investigation of the sparsity of the latent space generated by an RNN, depending on the task, the input\output dimension, and the latent space dimension. It may require some additional constraints or assumptions about Wzz and Wxz. But I don't see much room for this additional thorough investigation within the boundaries of this paper.

---

### Official Review · Reviewer_GAhp · 2022-10-24

**Confidence:** 4
**Correctness:** 3
**Technical Novelty And Significance:** 3
**Empirical Novelty And Significance:** 3
**Recommendation:** 5

**Clarity, Quality, Novelty And Reproducibility:**

Clarity:
The concept is clear.

Qaulity:
The idea is realtively novel but the task is simple.

Reproducibility:
Need some more details

**Strength And Weaknesses:**

Strength:
1.	It is interesting to find the connection between RNN with a sparse coder, which is then used for few-shot learning in speckle suppression and image deblurring tasks.
2.	It is surprising to obtain reasonably good performance by training the model on a single training image set.
3.	Clear explanation about the newly proposed perspective of data generation w.r.t. AEP.

Weakness:
1.	At Page 4, right after Eq.3, it is claimed that the sample size should be necessarily large. However, I fail to see clear connection between the sample size and the proposed RNN-based approach.
2.	Though RNN can be used as an unfolding of an iterative algorithm, it still requires the assumption that different columns in single patches are formulated as a sequence with clear order. However, this may not be true. At least a bi-directional RNN considering the order of columns along two complementary directions should be considered.
3.	The experiments are not comprehensive. In Table 1, the performance difference between RNN & RNN-RFN is not significant. A more detailed explanation should be provided. Also, the PSNR and SSIM are not clearly labeled (I assume only the bottom approaches have SSIM scores).
4.	What are the compared approaches? The author should at least give a reference for approaches listed in Table 1 and explain why the performance underperforms the compared approaches by a large margin.
5.	As mentioned above, the tasks used for evaluation are too simple. In addition to the simple cases, evaluation on a more challenging task (e.g., a more challenging noise situation added in image deblurring) should be performed.


**Summary Of The Paper:**

This paper targets the understanding of few-shot learning under the image reconstruction task. Specifically, it focuses on OCT speckle suppression and image deblurring, which use local information to clean the speckle/noise from the full image. The performance on the two tasks has successfully demonstrated the effectiveness of the approach though the tasks are kind of simple and may be of limited impact. Regarding the approach side, it first provides a theoretical analysis of few-shot learning from the perspective of asymptotic equipartition property and shows that there is a relatively small set (i.e., typical set) whose sample entropy is close to the true entropy and can represent the whole data distribution. Then, it proposes to connect RNN with sparse coding and treat the RNN as an unfolding of one iteration of the sparse coder. Finally, the RNN network is used in few-shot learning for related tasks.

**Summary Of The Review:**

Though detailed analyses are provided, the experiment is not quite promising, and it is hard to measure the impact of this approach. In addition, as mentioned above, further discussion is needed to determine whether the image patches can be formulated as sequences.

---

> ### Author Response · Authors · 2022-11-17
> **Reply to reviewer GAhp (part A)**
>
> We would like to thank you and the anonymous reviewers for the valuable comments and helpful suggestions. A revised version of the referenced manuscript has been submitted on-line. The responses to the reviewers’ comments are summarized as follows.
>
> Re: Summary of the paper, Strength and Weaknesses
>
> 1. Corrected. Theoretically, as now re-stated in paper: “as long as the sample size is larger or equal to the typical set size, we are guaranteed to have a generalization error that is as small as the training error. Otherwise, the upper bound on the generalization error depends on the ratio between the sample size and the input typical set size.”
> In other words, the sample size depends on the entropy of the data distribution (not necessarily large, or as large as the entire data distribution set). The proposed RNN approach serves as an example for few-shot learning, demonstrating the theoretical hypothesis according to which there exists a smaller set that is sufficient for generalization. Different architectures could be further explored in future work to verify the AEP theory for ML.
> We emphasize that, as stated in the paper: “The AEP tells us that there exists a relatively small group of training examples that would be sufﬁcient for generalization. However, the AEP property does not deﬁne this set, nor the correct coding, learning or prediction method. It just reassures us that there exists a set of the sort.”
> At this point we are not yet proposing a strategy for defining the typical set and\or proposing a learning strategy that relies only on the typical set. The RNN analysis as a sparse coder is not directly related with the AEP concepts.
>
> The theoretical part is closely related to the experimental part. To more formally explore this connection, we added Figure 9 and the corresponding text: ”As the focus of this study is to bridge between theory and application for few-shot learning, and to provide theoretical justifications, rather than to achieve state-of-the-art results, we verified the theoretical results described in Theorem 4 in practice for the image deblurring case. To this end, we trained the RNN network with different sample sizes while trying to keep a constant training error,  $\mathcal{L} \leq \Delta_{m}$. The training data consists of patches belonging to a single image (starfish). We consider the recovery error, denoted as $\hat{\mathcal{L}}(h_{\Psi})$,
> over the other non-training 9 images as an empirical approximation for the generalization error. Figure 9 shows the evolution of the recovery error with increasing sample size chosen uniformly in the range [1, 61752].Evidently, $\hat{\mathcal{L}}(h_{\Psi}) < \mathcal{L} $ while $\mathcal{L}\geq\Delta_m$ ,which implies $m<2^{mI(X;Y)}$. In this region $\hat{\mathcal{L}}(h_{\Psi})  \leq 1 - m 2^{-nI(X;Y)}(1-\Delta_m) \leq \Delta$ (see proof). Hence, empirically we can deduce ${I(X;Y)} \geq 0.33$.”
> (I replaced the notation for the training error here due to in the Latex  compiler)
>
> 2. Note that the patches-based RNN framework described in Section 5 considers each row in the patch as the time step input signal to the RNN (see analysis patch definition and the following explanation). Different columns in single patches are formulated as a sequence with clear defined order. Generally speaking, the use of RNN is less intuitive\popular for computer vision tasks, because the RNN is causal. It only “sees” in one direction (”backwards”). It may be possible also to consider a bi-directional RNN implementation.
> To our regret, due to the 9 pages limit we couldn’t add an illustration describing the analysis patch. The definition simply describes a patch, that moves along the image, pixel by pixel.  Each row of the patch is a time step input to the RNN.
>
> 3. Corrected. We have made some minor changes in section 6.1.
> As stated in the paper “Despite lower qualitative score, RFN generally suppresses noise better visually.” Indeed, only the proposed approaches have SSIM scores. RNN-RFN is described in Section 5. We now write: “We used the set of natural images provided by Romano et al. (2017), followed by the experiments conducted for the nonlocally centralized sparse representation (NCSR) algorithm (Dong et al.,2013).” The experiments are designed to match the image-deblurring experiments for state of the art image deblurring methods: “Table 1 presents the optimal PSNR and SSIM scores obtained for each image, and compares the PSNR scores to state-of-the-art image deblurring methods: FISTA (Beck & Teboulle, 2009b), NCSR (Dong et al., 2013) and RED (Romano et al., 2017).”

---

> ### Author Response · Authors · 2022-11-17
> **Reply to reviewer GAhp (part B)**
>
> 4. Corrected. In Section 6.1 we added the required references to the compared approaches in the table. As stated in this section: “Note that most deblurring methods require prior knowledge of the degradation process. Whereas the proposed approach requires only one example of the degraded image and its corresponding ground truth. The best scores were obtained by training with either one of the images: butterfly, boats, parrot, starfish and peppers. Hence, it is safe to assume that the patches in these images reliably represent the typical set associated with natural images to a certain degree… That said, at this stage, we are not suggesting that the quality scores of this framework are competitive with state-of-the-art denoisers. Currently, the main advantage is substantial speed. Tensorflow \citep{TF:2016} implementation converges in about 1700 iterations in an average of 14.29 seconds on a laptop GPU (NVIDIA GeForce GTX Ti 1650 with 4GB video memory) and 2.01 minutes on i-7 CPU. Pytorch \citep{Pytorch:2019} implementation training converges in 40 epochs on average in 30.73 seconds on a GPU NVIDIA GeForce RTX 2080 Ti, and 2.73 minutes on i-7 CPU. Inference takes $\sim$500 msec. For reference, RED-SD takes 15-20 minutes of processing for each image, on a CPU.”
>
> 5. As mentioned above the image deblurring experiment matches the experiments presented by previous methods’ papers: FISTA, RED and NCSR. To clarify that, we now write:
> “First, we follow the same Gaussian deblurring experiment performed in Romano et al. (2017), using the set of natural images provided by the authors.”
> We added Figure 6-8 and the corresponding text: “Figure 6 show the evolution of the PSNR scores for RNN and RNN-RFN with increasing additive noise variance $\sigma_n \in [0,10\sqrt{2}]$ . As the noise level is increased RFN suppresses noise better. Additional examples and training details are in Appendix D.1.”
> Of course, we agree that it is possible to experiment with more challenging settings and analyze the results. For the sake of brevity, we settled for one relatively simple experiment that is comparable with known methods.  Our purpose was solely to point out that initial numerical results demonstrate the viability and efficiency of the proposed algorithms on image deblurring problems. It is possible to extend the theoretical approach generally, and the specific few-shot learning framework to other applications and other tasks, out of the scope of this paper.
> Our hope was that the OCT experiment will demonstrate the applicability of the proposed approach to real-world problems, and the advantage of its efficiency for real-time applications. In medical imaging, speckle suppression is considered an important problem that has yet not been convincingly solved and there is still room for further research. Speckle noise significantly degrades OCT images and ultrasound images and can interfere with correct medical diagnosis. Supervised learning speckle suppression methods require ground truth data, which is often unavailable, or available with system mismatch. The proposed experiment may seem simple, yet it is of high significance. It shows that training given one image solely is possible, even when the test images have significantly different visual structure, and\or were acquired with a different system (different sampling rate and different resolution).  These experiments open the door for many more future research directions and practical implementations. As can be seen in [1]-[3] (below), normally de-speckling learning methods are far more complex and use hundreds, or even thousands of ground truth images for training. Also, most software-based de-speckling methods tend to smear the images.
>
> [1] Dong, Zhao, et al. "Optical coherence tomography image denoising using a generative adversarial network with speckle modulation." Journal of Biophotonics 13.4 (2020).
>
> [2] Ma, Yuhui et al. “Speckle Noise Reduction in Optical Coherence Tomography Images Based on Edge-Sensitive cGAN.” Biomedical optics express 9.11 (2018)
>
> [3] Shi, F., Cai, N., Gu, Y., Hu, D., Ma, Y., Chen, Y. and Chen, X., 2019. DeSpecNet: a CNN-based method for speckle reduction in retinal optical coherence tomography images. Physics in Medicine & Biology, 64(17), p.175010.
> Re Clarity, Quality, Novelty And Reproducibility:

---

> ### Author Response · Authors · 2022-11-17
> **Reply to reviewer GAhp (part C)**
>
>
> Re Quality: addressed above.
> Re Reproducibility: Corrected. We now write: “Our code will be available upon acceptance.” Unfortunately, we cannot publish an anonymous open-source code at this stage. The real OCT data is currently still confidential. Our request to open-source only the code has also been denied, since our organization is planning to use the code on an exclusive basis for upcoming projects. The RNN framework is quite straightforward to implement both in Pytorch or in Tensorflow. The images used for the image deblurring part are available in previous works [1]: https://github.com/google/RED.
> And for the theoretical results, clear explanations of any assumptions and a complete proof of the claims is included in the paper and the appendices.
>
> Re Summary Of The Review:
> The contributions are of great significance to learning theory and applications in general. As far as we know, the paper’s contributions – both on the theoretical side and the empirical side - are completely novel and of great significance to the ML community. We made all the necessary corrections advised by the reviewer to improve Clarity and Quality and replied to the reviewer’s comments regarding novelty and reproducibility. We hope that the reviewer can reconsider the paper for publication.

---

### Official Review · Reviewer_uFRZ · 2022-10-25

**Confidence:** 3
**Correctness:** 1
**Technical Novelty And Significance:** 2
**Empirical Novelty And Significance:** 2
**Recommendation:** 3

**Clarity, Quality, Novelty And Reproducibility:**

*Clarity
- In p.3, there are no definitions for g(x) and e(x).
- In p.3, n has multiple meanings for the superscript n for x^n and the number of dimensions of x, n, which is confusing.
- In Eq.1 and 2, x and y are not bold. Is it a typo? Or are they different from other bold x and y?
- In Eq.3, there is no definition for Delta.
- The character lambda is duplicately used in different meanings as the parameter in Eq.7 and eigenvalue.
- In Eq.15, there is no bias term compared to Eq.5. Are there any analyses or discussions with that?
- In Eq. 11 and 15, what can we interpret W and S as (like D is a dictionary)?
- How do the authors optimize W and S? What are training samples for z?
- Definition 1 is hard to follow. Providing some illustrations can help.

*Quality
- Please see the above comments.

*Novelty
- The proposed method might be novel.

*Reproducibility
- Code is unavailable.

**Strength And Weaknesses:**

*Strength
- Experimental results of the proposed method showed promising performance.

*Weaknesses
- The relationship between the theoretical part (Section 2) and the actual implementation part (Sections 3-5) is unclear. The theory of sparse coding itself might work for this instead of the discussion of AEP.
- Clarity is low.

**Summary Of The Paper:**

The authors provide some theoretical statements showing that a relatively small set that can empirically represent the input-output relationship for learning.
They then pointed out that sparse coding with iterative solutions can be modeled by RNN computation.
They applied the proposed method to few-shot learning.
Experimental results on image deblurring and optical coherence tomography (OCT) speckle suppression demonstrate that the proposed method performs well.

**Summary Of The Review:**

The relationship between the theoretical part and the actual implementation part is unclear. Clarity is low.

---

> ### Author Response · Authors · 2022-11-17
> **Reply to reviewer uFRZ (part A)**
>
> Reply summary of the paper, Strength And Weaknesses: Corrected.
> Note that, as stated in the introduction (main contributions): In this work we investigate the theoretical and empirical possibilities of few shot learning and the use of RNNs as a powerful platform given limited ground truth training data. In the first part we are basically justifying the promising performance of the implementation part. The RNN framework serves as an example where learning was successfully accomplished with limited data. The theoretical part (Section 1) states that based on the information-theoretical asymptotic equipartition property (AEP) (Cover & Thomas, 2006), there exists a relatively small set that can empirically represent the input-output data distribution for learning. Then in Sections 3-6, in light of the theoretical analysis, we promote the use of a compact RNN-based framework and demonstrate the applicability and efficiency for few-shot learning for natural image deblurring and optical coherence tomography (OCT) speckle suppression. We demonstrate the use of a single image training dataset, that generalizes well, as an analogue to universal source coding with a known dictionary. The method as well as the AEP learning can be applicable to other learning architectures as well as other applications where the signal can be processed locally, such as speech and audio, video, seismic imaging, MRI, ultrasound, natural language processing and more.
>
> In the end of Section 2 we clarify that “The AEP tells us that there exists a relatively small group of training examples that would be sufficient for generalization. However, the AEP property does not define this set, nor the correct coding, learning or prediction method. It just reassures us that there exists a set of the sort. How do we find the typical learning set? One option may be by predefined or learned dictionary coding: build a training set that represents the typical set, consisting of the most common structures, in a similar manner to universal source coding based on a known dictionary (Cover & Thomas, 2006).”
> At this point the problem of designed learning from the AEP is still an open question as stated in the conclusions: “Our future work will extend the AEP hypothesis to the automatic design of the AEP training dataset for unsupervised medical imaging learning tasks.”
> Nevertheless Sections 5-6, definitely demonstrate that in practice it is possible to successfully learn and generalize with a relatively small training set. As stated in Section 6.1.
>
> The experimental results are closely related to the theoretical part. To explore this connection more formally, we added Figure 9 and the corresponding text: ”As the focus of this study is to bridge between theory and application for few-shot learning, and to provide theoretical justifications, rather than to achieve state-of-the-art results, we verified the theoretical results described in Theorem 4 in practice for the image deblurring case. To this end, we trained the RNN network with different sample sizes while trying to keep a constant training error, $\mathcal{L}\leq\Delta_m$. The training data consists of patches belonging to a single image (starfish). We consider the recovery error, denoted as $\hat{\mathcal{L}}(h_{\Psi})$, over the other non-training 9 images as an empirical approximation for the generalization error. Figure 9 shows the evolution of the recovery error with increasing sample size uniformly chosen in the range [1,61752] and the training error. Evidently, $\hat{\mathcal{L}}(h_{\Psi}) < \mathcal{L} $ while $\mathcal{L}\geq\Delta_m$ ,which implies $m<2^{mI(X;Y)}$. In this region $\hat{\mathcal{L}}(h_{\Psi})  \leq 1 - m 2^{-nI(X;Y)}(1-\Delta_m) \leq \Delta$ (see proof). Hence, empirically we can deduce ${I(X;Y)} \geq 0.33$.”
> (I replaced the notation for the training error here due to a bug in the Latex compiler)
>
> We believe this work is of great significance and would enable a significant speed-up in learning tasks, not necessarily just inverse problems, or RNN-based frameworks, as stated in our conclusions: “Future work could extend the proposed work to other tasks and applications in other domains.”
>
> Unfortunately, we had to redact\move to the supplementary materials most of the sparse representations’ theory due to the 9-pages limit.

---

> ### Author Response · Authors · 2022-11-17
> **Reply to reviewer uFRZ (part B)**
>
> Re:Clarity, Quality, Novelty And Reproducibility:
> Corrected.
> Comments:
> *We now write in the beginning of the INFORMATION THEORETIC PERSPECTIVE: CONNECTION TO AEP  part: “Hereafter, we use the notation $x^n$ to denote a sequence $x_1,x_2,...,x_n$.” The superscript ^n refers to the dimension of x. In information theory the notation for the block sequence of length n (n symbols) is commonly $x^n$. For a vector $\mathbf{x}$ that is the dimension of x. Note that n defines the length of the vector u. d is the length of the vector y. x^n and $\mathbf{x}$ are interchangeably used to denote the same variable. As stated in the proofs section “Throughout the proofs we sometimes omit the superscript $^n$ for simplicity.”
> * We added $\Delta_m \leq \Delta \leq 1$ to equations (3) and (4). Delta is defined by equation 3 as the variable that connects between the sample size (relative to the size of typical set and Delta_m) and upper bounds the generalization error.
> * Notations for lambda, and $\lambda_max$ were initially borrowed from (Elad, 2010). We replaced $\lambda_max$ with $\alpha_max$.
> * The bias term in equation (15) is \beta. Depending on the implementation, it can be a single constant or a vector of constants, but for the sake of brevity we rather not dive into these details in the scope of this paper.
> * As can be derived from equation (10) W can be interpreted as $\frac{1}{c}\mathbf{D}^T$, and  S can be interpreted as $(I-\frac{1}{c}\mathbf{D}^T\mathbf{D})$ where I is the identity matrix.
> We added a second equality to eq. (10) to make that clearer to the reader.
> * Note that equation (11) describes learned-ISTA (Gregor & Lecun, 2012). As stated in the paper S and W are learned over a known set of training samples
> $\{\mathbf{y}_i,\mathbf{z}_i\}_{i=1}^m$.
> *We corrected definition 1 to make it easier to follow. To our regret, due to the 9 pages limit we couldn’t add a figure describing the analysis patch. The definition simply describes a patch, that moves along the image, pixel by pixel.  Each row of the patch is a time step input to the RNN.
>
> Re Novelty: As far as we know, both the theoretical analysis and the experimental study cases presented in the paper are completely novel (except any part that is explicitly cited otherwise). This includes: our insights into the asymptotic equipartition property (AEP) (Shannon, 1948) in the context of machine learning and its potential ramiﬁcations for few-shot learning, the theoretical guarantees for reliable learning under the information-theoretic AEP, and for the generalization error with respect to the sample size, application of recurrent neural net (RNN) framework for few shot learning for image deblurring and OCT speckle suppression, RNN-RFN proposed reduced-entropy algorithm for few-shot learning; and the mathematical intuition for the RNN as an approximation of a sparse coding solver.
>
> As stated in the paper, the RNN framework is employed here to demonstrate one possible outcome of the AEP theory: “The setting described in this subsection has been previously employed for various applications. Biswas et al. (2018) and Pereg et al. (2020a) used a similar framework to facilitate automatic velocity analysis, where they used a portion of the acquired data for training and let the system infer the rest of the missing velocities. Pereg et al. (2020b) used a similar framework to perform seismic inversion with synthetic training data. Here, we expand and elaborate its application, while connecting it to the theoretical intuition in Section 2.”
>
> Re Reproducibility: Corrected. We now write: “Our code will be available upon acceptance.” Unfortunately, we cannot publish an anonymous open-source code at this stage. The real OCT data are currently still confidential. Our request to open-source only the code has also been denied, since our organization is planning to use the code on an exclusive basis for upcoming projects. The RNN framework is quite straightforward to implement both in Pytorch or in Tensorflow. The images used for the image deblurring part are available in previous works [1]: https://github.com/google/RED.
> And for the theoretical results, clear explanations of any assumptions and a complete proof of the claims is included in the paper and the appendices.

---

> ### Author Response · Authors · 2022-11-17
> **Reply to reviewer uFRZ (part C)**
>
>
> Re Summary: The contributions are of great significance to learning theory and applications in general. As far as we know, the paper’s contributions – both on the theoretical side and the empirical side - are completely novel and of great significance to the ML community.
> As stated by reviewer GAhp “Strength: 1. It is interesting to find the connection between RNN with a sparse coder, which is then used for few-shot learning in speckle suppression and image deblurring tasks. 2. It is surprising to obtain reasonably good performance by training the model on a single training image set. 3. Clear explanation about the newly proposed perspective of data generation w.r.t. AEP.”
>
> We made all the necessary corrections advised by the reviewer to improve clarity and quality and replied to the reviewer’s comments regarding novelty and reproducibility. We hope that the reviewer could reconsider the paper for publication.  We would like to thank you and the anonymous reviewers for the valuable comments and helpful suggestions. A revised version of the referenced manuscript has been submitted on-line. The responses to the reviewers’ comments are summarized as follows.

---

### Official Review · Reviewer_WSnj · 2022-10-25

**Confidence:** 3
**Correctness:** 2
**Technical Novelty And Significance:** 2
**Empirical Novelty And Significance:** 1
**Recommendation:** 3

**Clarity, Quality, Novelty And Reproducibility:**

The paper is unnecessarily complicated to read and doesn't focus on it's topic. Sections 3 and 4 give a lot of details about sparse coding that do not add much to the proposed approach.

**Strength And Weaknesses:**

The empirical evaluation of the paper is very very limited. If the authors want to make a convincing argument they will have to use more standard datasets and make comparisons to better approaches. The RNN model performs very close to the RNN-RFN according to Table 1. I wouldn't be surprised if dropout or l2 regularization would push the performance of a basic RNN beyond the other approaches.

I haven't fully understood the mathematical results, but they seem overly complicated without adding much to the proposed approach (I spent quite some time going through them).

**Summary Of The Paper:**

First, the authors make some proofs (whose correctness I haven't fully verified) based on  the asymptotic equipartition property that show there exists a small set that can represent the whole data distribution. Then, the authors make some connections between RNNs and sparse coding. Finally, authors test RNNs and RNNs with receptive field normalization in some ad-hoc tasks without implementing proper baselines.

**Summary Of The Review:**

The paper needs to have a clear focus and structure. The theoretical results don't seem insightful. On top of that, the empirical evaluation of the results is far from enough to make any serious claims about the approach.

---

> ### Author Response · Authors · 2022-11-17
> **reply to reviewer WSnj (part A)**
>
> As stated by the reviewer the purpose of the paper is to emphasize the existence of a small dataset that can represent the entire data distribution. In addition, the paper highlights the connection between RNNs and sparse coding. The experimental results demonstrate the applicability of the theoretical statements, comparing to other methods, showing qualitative and quantitative evaluation of the proposed point of view.
> To the best of our knowledge our image deblurring baseline is of state-of-art deblurring methods, as stated in the paper, such as:  Regularization by denoising (RED) [1], Nonlocally centralized sparse representation (NCSR) [2]. The first experiment follows the exact deblurring experiment presented by previous methods’ papers: FISTA, RED and NCSR. To clarify that, we now write: “First, we follow the same Gaussian deblurring experiment performed in Romano et al. (2017), using the set of natural images provided by the authors.”
>
> The OCT experiments demonstrate the applicability of the framework to a real-world problem. Although many handcrafted and learned algorithms have been developed for speckle suppression, it remains a major obstacle for the clinical use of coherent imaging modalities. There exists no validated method to perform speckle suppression. Our experiments make two significant steps beyond previous OCT speckle suppression studies: First, we use experimentally obtained ground-truth. Second, we investigate the generalization of the learned RNN to different tissue types and acquisition settings. Supervised speckle suppression often uses temporally averaged tomograms as ground-truth, which is intrinsically flawed due to sample motion and resulting blurring of tissue structure. Our ground-truth, in comparison, defines speckle-suppressed tomograms with preserved spatial resolution, which makes speckle suppression both more challenging and more impactful. Furthermore, the fact that the RNN can be trained on a given tissue type and achieves convincing performance on different tissues is critical for its practical application. It also makes an important connection back to the AEP, suggesting that the training data contained the typical set and hence guarantees the successful inference of other tissue types.
> There is obviously a lot of room for further exploration, but we trust that the condensed presentation of these OCT results in the paper offer compelling evidence for the potential of RNN and few-shot learning in practical applications.
>
> Re Strength and weaknesses:
> The datasets used for image deblurring is quite standard for this task. FISTA [3], RED [1,4] and NCSR [2] are state of the arts results for image deblurring.
> The OCT dataset is indeed an in-house dataset, but it is common to use non-standard datasets in the field. The ground truth images used for this example for evaluation are of the best quality in the field. Qualitative measures (PSNR) and visual examples illustrate the applicability of the proposed approach. We believe that for the sake of proof of concept, having 2 case studies, with 2 optional implementations, and comparing to conventional methods, sufficiently demonstrates the robustness of the method and its potential applicability. We leave other extensions to future research and industrial R&D.
>
> The experimental results are closely related to the mathematical results. To explore this connection more formally, we added Figure 9 and the corresponding text: ”As the focus of this study is to bridge between theory and application for few-shot learning, and to provide theoretical justifications, rather than to achieve state-of-the-art results, we verified the theoretical results described in Theorem 4 in practice for the image deblurring case. To this end, we trained the RNN network with varying sample sizes while trying to keep a constant training error, $\mathcal{L} \leq \Delta_{m}$. The training data consists of patches belonging to a single image (starfish). We consider the recovery error, denoted as $\hat{\mathcal{L}}(h_{\Psi})$,
> over the other non-training 9 images as an empirical approximation for the generalization error. Figure 9 shows the evolution of the recovery error with increasing sample size chosen uniformly in the range [1, 61752].Evidently, $\hat{\mathcal{L}}(h_{\Psi}) < \mathcal{L} $ while $\mathcal{L}\geq\Delta_m$ ,which implies $m<2^{mI(X;Y)}$. In this region $\hat{\mathcal{L}}(h_{\Psi})  \leq 1 - m 2^{-nI(X;Y)}(1-\Delta_m) \leq \Delta$ (see proof). Hence, empirically we can deduce ${I(X;Y)} \geq 0.33$.”
> (I replaced the notation for the training error here due to in the Latex  compiler)

---

> ### Author Response · Authors · 2022-11-17
> **Reply to reviewer WSnj (part B)**
>
> We also added Figure 6-8 and the corresponding text: “Figure 6 show the evolution of the PSNR scores for RNN and RNN-RFN with increasing additive noise variance $\sigma_n \in [0,10\sqrt{2}]$ . As the noise level is increased RFN suppresses noise better. Additional examples and training details are in Appendix D.1.”
> Of course, we agree that it is possible to experiment with more challenging settings and analyze the results. For the sake of brevity, we settled for one relatively simple experiment that is comparable with known methods.  Our purpose was solely to point out that initial numerical results demonstrate the viability and efficiency of the proposed point of view on image deblurring problems. It is possible to extend the theoretical approach generally, and the specific few-shot learning framework to other applications and other tasks, out of the scope of this paper.
>
> Unfortunately, so far additional upgrades such as l2 regularization, weight decay and dropout did not yet yield substantial improvement in the results for the specific case studies presented in the paper. Of course, we agree that there can be many more practical extensions as we write in the image deblurring section: “That said, at this stage, we are not suggesting that the quality scores of this framework are competitive with state-of-the-art denoisers. Currently, the main advantage is substantial speed.”
> We agree that the qualitative scores for the RNN are very close to the RNN-RFN, but as mentioned in the paper: “Despite lower qualitative score, RFN generally suppresses noise
> better visually.” We added Figures 6-8 and the corresponding text to demonstrate this property.
> Future work may include other extensions, such as different architectures, specific implementation details, other task, or additional datasets.  For the sake of brevity, we rather not further dive into these issues in the scope of this paper.
> Furthermore, we would like to emphasize that as stated in the paper: “Training of the proposed framework is extremely time efficient. Training takes about 1-30 seconds on a GPU workstation and a few minutes on a CPU workstation (2-4 minutes),
> and thus does not require expensive computational resources.”
> One of the main takeout of the paper is that there is high potential for improving time efficiency for ML, which is currently a major obstacle in the field. We are providing both the theoretical and practical foundations for our statements, which we hoped would not be easily overlooked.
>
> The mathematical results are described in a simple manner designed to be clear to any reader with minimal mathematical background. We have intentionally written this section so that it would require minimal background in information theory. The reviewer stated that this section does not add much to the proposed approach, yet as stated by the reviewer: “the authors make some proofs that show there exists a small set that can represent the whole data distribution”. In other words, we first establish a theoretical basis for few-shot learning, and then show how it can be practically achieved with a specific RNN framework, for example. That is, the focus of the paper as defined in the title “Less is more: Rethinking Few-Shot Learning and RNNs”.
>
> Re Clarity, Quality:
> Reply: Corrected. We added some comments and figures to section 6.1 to comply with the reviewers’ requirements.
> In our opinion, the background on RNNs, sparse coding, and iterative thresholding is necessary for the full understating of the paper to all readers, without requiring reading of previous works. We did our best to make the paper as brief, simple and clear as possible, independently comprehensible for any reader without requiring previous reading of other papers. In Sections 3 and 4, we provide all necessary information on the problem statement, objectives, hypothesis, strategy or method, and data.
> As stated in the abstract and in Section 4: We illuminate a possible optimization mechanism behind RNNs. We observe that an RNN can be viewed as a sparse solver starting from an initial condition based on the previous time step. The proposed interpretation can be viewed as an intuitive explanation for the mathematical functionality behind the popularity and success of RNNs in solving practical problems.
> We were not at all suggesting that this is the sole focus of the paper.
> We would like to emphasize that as stated in the paper the main goal of the paper is *not* to provide a specific very limited empirical method and provide examples. We are presenting general learning principles that could be applied to other applications and architecture in the future.
>
> Note that Reviewer uFRZ is suggesting a contradiction to the above reviewer’s comment (that sections 3-4 are redundant): “The theory of sparse coding itself might work for this instead of the discussion of AEP.”

---

> ### Author Response · Authors · 2022-11-17
> **Reply to reviewer WSnj (part C)**
>
>
> Re Summary:
> In our humble opinion and as stated by reviewer uFRZ: “Strength: Experimental results of the proposed method showed promising performance.”
> The paper’s claims are verified and supported by rigorous mathematical theory and experimental results. The contributions are of great significance to learning theory and applications in general. As far as we know, the paper’s contributions – both on the theoretical side and the empirical side - are completely novel and of great significance to the ML community.
>
> We made all the necessary corrections advised by the reviewers to improve clarity and quality and replied to the reviewer’s questions. It is our hope that the reviewer could reconsider the paper for publication.
>
> We would like to thank you and the anonymous reviewers for the valuable comments and
> helpful suggestions. A revised version of the referenced manuscript has been submitted
> on-line. The responses to the reviewers’ comments are summarized as follows.
>
> [1] Yaniv Romano, Michael Elad, and Peyman Milanfar. The little engine that could: Regularization by denoising (RED). SIAM Journal on Imaging Sciences, 10(4):1804–1844, 2017.
>
> [2] Weisheng Dong, Lei Zhang, Guangming Shi, and Xin Li. Nonlocally centralized sparse representation for image restoration. IEEE Transactions on Image Processing, 22(4):1620–1630, 2013.
>
> [3] Amir Beck and Marc Teboulle. Fast gradient-based algorithms for constrained total variation image denoising and deblurring problems. IEEE transactions on image processing, 18(11):2419–2434,2009b.
>
> [4] Cohen, R., Elad, M. and Milanfar, P., 2021. Regularization by denoising via fixed-point projection (RED-PRO). SIAM Journal on Imaging Sciences, 14(3), pp.1374-1406.

---

### Author Response · Authors · 2022-11-17
**Response to review**

Re: 	“Less is More: Rethinking Few-Shot Learning and Recurrent Neural Nets”

Dear Area Chair,

We would like to thank you and the anonymous reviewers for the valuable comments and
helpful suggestions. A revised version of the referenced manuscript has been submitted
on-line. The responses to the reviewers’ comments are summarized as follows.


Sincerely

Anonymous Authors

---

### Author Response · Authors · 2022-11-18
**Paper update**

An update revised version of the manuscript has been submitted on-line.
Figure 9 and the corresponding text have been replaced.
Further clarification on equation (11) has been added ("Note that W and S are no longer constrained D").

---

### Decision · Program_Chairs · 2023-01-20

**Decision:**

Reject

**Justification For Why Not Higher Score:**

The reviewers unanimously rate the current version of the manuscript below the accept line, so is the decision.

**Justification For Why Not Lower Score:**

N/A

**Metareview: Summary, Strengths And Weaknesses:**

This manuscript studies few-shot learning under the image reconstruction task. Specifically, it focuses on OCT speckle suppression and image deblurring, which use local information to clean the speckle/noise from the full image. It first provides a theoretical analysis of few-shot learning from the perspective of asymptotic equipartition property and shows that there is a relatively small set (i.e., typical set) whose sample entropy is close to the true entropy and can represent the whole data distribution. Then, it proposes to connect RNN with sparse coding and treat the RNN as an unfolding of one iteration of the sparse coder. Finally, the RNN network is used in few-shot learning for related tasks.

Although it is interesting to see the connection of RNN with sparse coding iterative algorithms, the reviewers have main concerns on the evaluation of the methodology and clarity of the logic flow. The reviewers unanimously rate the current version of the manuscript below the accept line, so is the decision.